

# Expanding global commodities trade and consumption place the world's primates at risk of extinction

Alejandro Estrada[1], Paul A. Garber[2] and Abhishek Chaudhary[3]

[1] National Autonomous University of Mexico, Institute of Biology, Mexico City, Mexico
[2] Department of Anthropology, University of Illinois at Urbana-Champaign, Urbana, IL, USA
[3] Department of Civil Engineering, Indian Institute of Technology Kanpur, Kanpur, India

## ABSTRACT

As a consequence of recent human activities. populations of approximately 75% of the world's primates are in decline, and more than 60% of species ($n$ = 512) are threatened with extinction. Major anthropogenic pressures on primate persistence include the widespread loss and degradation of natural habitats caused by the expansion of industrial agriculture, pastureland for cattle, logging, mining, and fossil fuel extraction. This is the result of growing global market demands for agricultural and nonagricultural commodities. Here, we profile the effects of international trade of forest-risk agricultural and nonagricultural commodities, namely soybean, oil palm, natural rubber, beef, forestry products, fossil fuels, metals, minerals, and gemstones on habitat conversion in the Neotropics, Africa, and South and Southeast Asia. Total estimated forest loss for these regions between 2001 and 2017 was *ca* 179 million ha. The average percent of commodity-driven permanent deforestation for the period 2001–2015 was highest in Southeast Asia (47%) followed by the Neotropics (26%), South Asia (26%), and Africa (7%). Commodities exports increased significantly between 2000 and 2016 in all primate range regions leading to the widespread conversion of forested land to agricultural fields and an increase in natural resource extraction. In 2016, US $1.1 trillion of natural-resource commodities were traded by countries in primate range regions. The Neotropics accounted for 41% of the total value of these exports, Southeast Asia for 27%, Africa 21%, and South Asia 11%. Major commodity exporters in 2016 were Brazil, India, Indonesia, Malaysia and South Africa, countries of high primate diversity and endemism. Among the top 10 importers were China, the US, Japan, and Switzerland. Primate range countries lag far behind importer nations in food security and gross domestic product per capita, suggesting that trade and commodity-driven land-use have done little to generate wealth and well-being in primate habitat countries. Modeling of land-use and projected extinction of primate species by 2050 and 2100 under a business as usual scenario for 61 primate range countries indicate that each country is expected to see a significant increase in the number of species threatened with extinction. To mitigate this impending crisis, we advocate the "greening" of trade, a global shift toward a low-meat diet, reduced consumption of oil seed, diminished use of tropical timber, fossil fuels, metals, minerals, and gemstones from the tropics, accompanied by a stronger and sustained global resolve to regulate and reverse the negative impacts of growing unsustainable global demands and commodity trade on income inequality, and the destruction of primates and their habitats.

Corresponding authors
Alejandro Estrada,
aestradaprimates@gmail.com
Paul A. Garber,
p-garber@illinois.edu

## INTRODUCTION

International commodities trade has become the backbone of many of the world's economies. Each year, trade in just a handful of agricultural and nonfood commodities such as soy, palm oil, and minerals drive billions of dollars of investment in both producer and consumer nations (*Henders, Persson & Kastner, 2015*; *Henders et al., 2018*; *MacDonald et al., 2015*). The volume of food and nonfood commodities traded globally has increased by over 60% since the turn of the century, with four trillion US dollars of natural resource commodities traded in 2016 (https://resourcetrade.earth/). The trade of these commodities has had a direct impact on land-use change and environmental and social policies in exporting countries (*Henders, Persson & Kastner, 2015*; *Henders et al., 2018*), and in many instances these policies, driven by the over-consumption of a few major importing countries, have resulted in large-scale deforestation, the poisoning of soil, streams, and rivers, and transformed biodiverse landscapes into monocultures (*Gardner et al., 2018*; *MacDonald et al., 2015*; *Moran, Petersone & Verones, 2016*). The demand for agricultural products and other commodities in importing countries is increasingly being met by global supply chains associated with multinational corporations rather than local producers (*Yu, Feng & Hubacek, 2013*), a process that weakens the ability of exporting countries to sustainably manage their own natural resources (*Meyfroidt et al., 2013*; *Lambin & Meyfroidt, 2011*). Thus, the environmental costs of production have been disproportionately borne by exporting nations, and both exporting and importing countries have become ever more dependent on external and distant resources to satisfy their food and natural resource security needs (*Fader et al., 2011*, *2013*; *MacDonald et al., 2015*). Global market demands for forest-risk commodities (e.g., soybeans, palm oil, hardwoods, and other commodities that contribute significantly to the conversion of biodiverse forests into monocultures or highly degraded habitats) have resulted in a process of rapid and widespread industry-driven deforestation, negatively impacting tropical biodiversity, subdividing single large or continuous wild animal populations into small and isolated subpopulations, reducing habitat suitability and gene flow, and limiting the area available for species distribution and population persistence (*Chaudhary & Brooks, 2018*; *Henders, Persson & Kastner, 2015*; *Henders et al., 2018*; *Wich et al., 2014*; *Estrada et al., 2017*, *2018*; *Li et al., 2018*). Today, commercial agriculture at local and global scales is the most significant driver of deforestation worldwide (*Hosonuma et al., 2012*).

Clearly, trade is a vital component of food security and all countries are dependent on the global food trade. Approximately 25% of all the food produced for human consumption crosses international frontiers (*D'Odorico et al., 2014*). Trade helps food security by providing a safety measure against oscillations in domestic food supply and by stabilizing prices (*World Bank, 2015*). Yet, trade dependence also means that all countries
are vulnerable to shifts in the trade policies of food-exporting nations, and to the disruption of supply chains associated with political instability, catastrophic weather events, trade wars, and export bans (*Rudolff, 2015*; *Fuchs et al., 2019*). Because the international trade of commodities is widespread and growing, and land conversion permanently alters natural habitats, effective management plans and conservation initiatives require a comprehensive understanding of the local, regional, and global impacts of consumption on biodiversity and environmental sustainability (*Chaudhary & Brooks, 2017*; *Chaudhary & Mooers, 2018*).

A major goal of this manuscript, is to examine the environmental impacts of the international trade of essential agricultural and nonagricultural commodities (e.g., trade flows) on primate habitats and population persistence in the major regions of the world that harbor wild primates (prosimians, tarsiers, monkeys, and apes): the Neotropics, Africa (for the purposes of this manuscript, we include Madagascar as part of Africa), South Asia, and Southeast Asia. Nonhuman primates are our closest biological relatives, represent the third most specious Order of Mammals, and play an essential role in the maintenance and regeneration of the tropical and temperate ecosystems they inhabit (*Estrada et al., 2017*). Current information indicates the existence of 512 primate species in 79 genera distributed in 91 countries. Alarmingly, the populations of about 75% of all primate species are declining and more than 60% of the species are now threatened with extinction (*IUCN, 2019*). Major anthropogenic pressures on primate population persistence are land cover changes caused by industrial agriculture, the expansion of pasture for cattle ranching, and increasing logging, mining, and fossil fuel extraction. These activities result in habitat loss and fragmentation, and the disruption of natural ecosystems caused by the building of dams and mega-dams, and the expansion of road and rail networks for resource extraction, along with the colonization of frontier forests, bushmeat hunting, the illegal trade of primates as pets and for their body parts, and the spread of human and domestic animal-borne infectious diseases. These drivers commonly act in synergy, exacerbating primate habitat loss (including in protected areas) and population decline (*Estrada et al., 2017*, *2018*). In most primate habitat countries, these pressures arise in the context of an expanding human population with low levels of socioeconomic development, income inequality, political instability, weak governance, and the ongoing export trade of agricultural and nonfood commodities to international markets (*Estrada et al., 2017*, *2018*; *Li et al., 2018*). Therefore, a second goal of our manuscript is to document the global food security index (FSI) and the gross domestic product per capita (GDPPC) for exporter and importer countries and discuss their relationship to consumer nation driven international trade, social well-being, and primate conservation.

## SURVEY METHODOLOGY

We collected and integrated information from several international databases. First, using information from the Global Forest Watch database for the period 2001–2017, we profiled forest loss (loss of >30% canopy cover) and commodity-driven deforestation (2001–2015) for primate-range countries. Commodity driven deforestation refers to permanent conversion and total clearing of forest to nonforest land for the purpose of agricultural

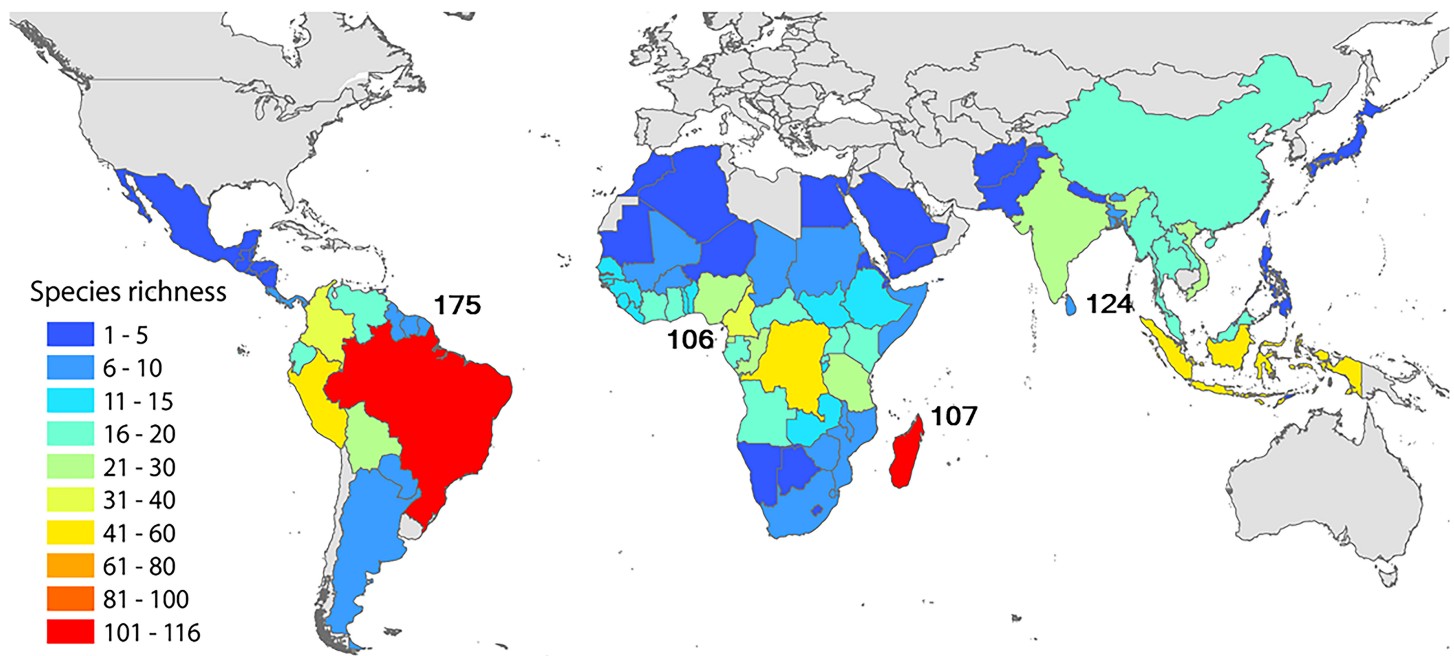

**Figure 1** Numbers in black indicate species richness in the main regions where primates are naturally found: the Neotropics, Africa (mainland Africa and Madagascar), South Asia, and Southeast Asia. The country colors indicate the number of species in each country. In Africa, Madagascar stands out with its rich and endemic primate fauna (the lemurs). The range of *Papio hamadryas* extends from mainland Africa into Asia with small populations present in Saudi Arabia and Yemen.

(e.g., oil palm, soybeans, natural rubber; *Henders, Persson & Kastner, 2015*; *Curtis et al., 2018*) or cattle (beef) production. The extraction of other commodities, such as land-based fossil fuels and ores, also results in forest loss and degradation due to the clearing of vegetation, and in the pollution of air, soil, and water (*Alvarez-Berríos & Mitchell Aide, 2015*; *Asner et al., 2013*; *Global Forest Watch, 2018*). Terminology used here for forest loss and commodity-driven deforestation is from *Global Forest Watch (2018)* and from *Curtis et al. (2018)*. Commodity production was profiled for 20 primate-range countries in the Neotropics, 35 in Africa, six in South Asia, and 14 in Southeast Asia (see Fig. 1).

Second, using Chatham House's international trade database, resoucetrade.Earth, we documented, for most countries in each primate-range region, the trade exports in 2016 of four agricultural commodities that contribute to forest fragmentation, degradation and habitat loss: soybeans, oil palm, natural rubber, and beef products (*Ahrends et al., 2015*; *Curtis et al., 2018*; *Henders, Persson & Kastner, 2015*), and four nonagricultural forestry products: fossil fuels, metals, minerals, and gemstones. We selected these eight commodities because they are among the most commonly traded worldwide. We used information from the year 2016 because it represents the most complete and up-to-date data set (*Beckman et al., 2017*; *Clapp, 2015*; *European Commission, 2013*; *Food and Agriculture Organization of the United Nations (FAO), 2018a*, *2018b*; *MacDonald et al., 2015*; *resourcetrade.Earth, 2018*). For a list of all commodities examined on resourcetrade. Earth see Text S1. Given the negative effects of these commodities on natural ecosystems,

we identified the countries that were the major exporters in each primate-range region in 2016, as well as the consumer nations that were the major importers of these commodities. Commodities terminology follows UNComtrade (https://comtrade.un.org) and resourcetrade.Earth (https://resourcetrade.earth/).

Third, agricultural and nonagricultural trade is principally an economic exchange, where food or nonfood products are provided by exporting nations to importing states in exchange for revenue (*Anderson, 2010*). This exchange can be examined in monetary value (US$) or mass traded (e.g., tonnage). We used monetary value as increases in value are strongly correlated with increases in the volume of commodities sold to importing countries as well as with the revenues generated by exporting countries (*MacDonald et al., 2015*). Value also can be viewed as a general measure of the footprint of importing countries on land-use practices in primate-range countries. Value similarly allows for a direct assessment of the magnitude of trade flows and can be correlated with other economic indicators such as GDP (*MacDonald et al., 2015*). With this in mind, we estimated the total revenue generated by the trade of commodities in 2016 by countries in each of the primate-range regions and compared this to the revenue reported for the year 2000.

We also assessed trends in the expansion of land area used to cultivate soy, oil palm, and natural rubber trees between 1960 and 2016 in the Neotropics, Africa, South Asia, and Southeast Asia, using information from FAOSTATS (Statistical division of the Food and Agriculture Organization of the UN; http://www.fao.org/faostat/en/#compare). Similarly, we analyzed growth in the number of cattle produced between 1960 and 2016 in primate-range regions, based on data available in the same database, and compared this to the growth in exports of beef by primate-range nations. Complementary information on commodity trade was obtained from the International Trade Centre (http://www.intracen.org/itc/market-info-tools/trade-statistics/), the UNComtrade database (https://comtrade.un.org), and the FAO trade database (http://www.fao.org/faostat/en/#compare). Because the international trade in agricultural products influences human food security across primate-range countries and regions, we examined the global FSI 2018 of The Economist Intelligence Unit Limited (https://foodsecurityindex.eiu.com/) for each primate-range commodity exporter country, and for importer nations. The FSI defines food security as the state in which people at all times have physical, social, and economic access to sufficient and nutritious food that meets their dietary needs for a healthy and active life. This framework is based on the internationally accepted definition established at the 1996 World Food Summit (http://www.fao.org/WFS/). The FSI uses three central pillars of food security—affordability, availability, and quality and safety, and ranges from 0 (lowest food security or highest food insecurity) to 100 (highest food security; see Text S1). We obtained, from the World Bank, GDPPC values for 2017 for primate-range exporting countries and for importer countries (https://data.worldbank.org/indicator/NY.GDP.PCAP.CD) and used these as indicators of the economic standing of primate-range commodity exporting countries compared to countries importing these products. Each of the agencies that we used as sources of data specify in their portals the constraints of the data they present.

We consider that, although in some cases the numbers reported may vary in their level of accuracy, the patterns and trends within and between each country or region are reliable with high confidence.

To illustrate international trade flows of commodities exported by primate-range regions and countries we developed Sankey flow diagrams by using the open access software SankeyMatic accessible in http://sankeymatic.com/build/. In these diagrams we used the accumulated value in $US as a general indicator of the footprint of importing nations. Important to consider in this assessment is the magnitude of the revenue accrued by each country that exports commodities.

## Modeling the current and future (2050 and 2100) risk of primate extinctions

We calculated the number of primate species currently threatened with extinction in each country, and compared this to the expected number of additional primate species in each country that will be threatened with extinction by 2050 and 2100 as a result of land use changes associated with forest-risk trade commodities. We examined six alternative scenarios. We used the approach presented by *Chaudhary & Mooers (2018)* that links the countryside species-area relationship model (*Chaudhary & Brooks, 2017*) with the current (2016) and future (2050 and 2100) global gridded maps generated by the six RCP-SSP combination scenarios (RCP 2.6 SSP-1, RCP 4.5 SSP-2, RCP 7.0 SSP-3, RCP 3.4 SSP-4, RCP 6.0 SSP-4, and RCP 8.5 SSP-5), available from the most recent land use harmonization (LUH2) dataset (http://luh.umd.edu/data.shtml: Table S9). We also allocated the total number of primate species threatened with extinction to 10 individual human land use types in the LUH2 database, taking into account each species' ability to utilize each of these altered habitats (*IUCN, 2015*) and to identify the major land use drivers threatening primate species in each country (*Chaudhary & Mooers, 2018*). See Text S1 for details on the methodology used.

We are aware that based on biogeography, differences in climate and topography, and historical patterns of economic development, land conversion, and political control, subregional or local variation in species richness within a country or among neighboring countries exists and therefore the use of a regional approach to primate conservation can mask fine-grained cause and effect relationships between commodity production and primate diversity. For example, in the Neotropics two neighboring countries, Brazil and Argentina differ by a factor of 20 in their number of primate species. Although global commodities trade in each country is having a highly negative effect on ecosystem's health and biodiversity, the number of primate species impacted in Brazil is significantly greater than in Argentina. Similarly, different parts of the same country may vary considerably in both primate diversity and the environmental affects of commodities trade. Bearing this in mind, and given the recent expansion of international commodity trade throughout primate-range countries, we feel that the data we present are best considered indicators of general trends at the global and regional scales.

**Table 1** Deforestation per year (loss of >30% tree-canopy cover) based on remote sensing for the period 2001–2017 for the four primate range regions under consideration.

| Region (n = countries) | Tree cover loss 2001–2017 (million ha) | Rate (n = 17 years) (million ha/year) | Average % commodity-driven deforestation per region 2001–2015 |
|---|---|---|---|
| Neotropics (n = 20) | 83.5 | 4.91 | 26 |
| Southeast Asia (n = 12) | 54.3 | 3.19 | 47 |
| Africa (n = 35) | 38.5 | 2.29 | 7 |
| South Asia (n = 7) | 1.9 | 0.11 | 26 |
| Total (2001–2017) | 178.8 | Four regions 10.52 | |

Notes:
Also shown is the rate of forest loss for each region and the average percent of permanent forest loss caused by commodity-driven forest loss for 2001–2015. Shown at the bottom is the total loss in millions of ha for the four regions. Regions are ranked by the magnitude of tree-cover loss between 2001 and 2017.
Based on remote sensing, Global Forest Watch (GWF) has classified five principal causes of forest loss in the world: forestry (26%); shifting agriculture (24%); wildfire (23%); commodity-driven deforestation and urbanization (27%). The last two are consider as drivers of permanent deforestation. Deforestation is measured by GFW as loss of >30% tree canopy cover.

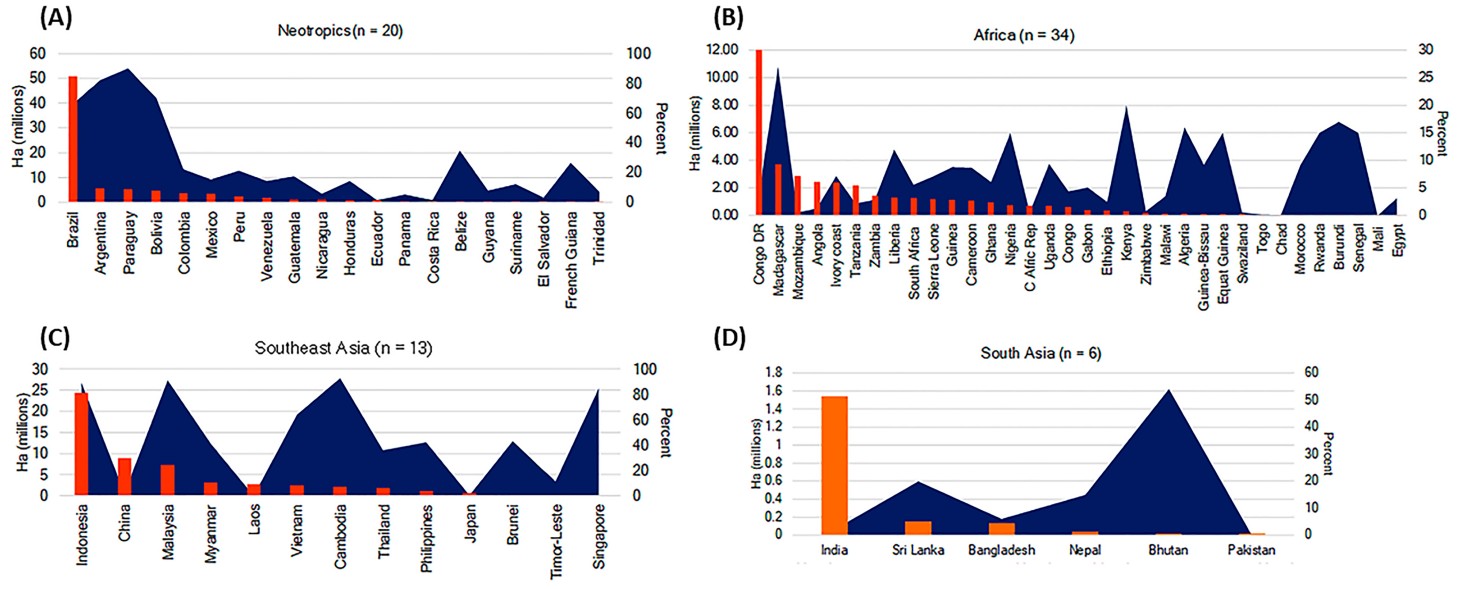

**Figure 2** Forest loss (reduction in >30% tree canopy cover) for 2001–2017 (orange bars) and percent of commodity-driven forest loss (permanent deforestation) for 2001–2015 (blue area). Primate range countries in (A) the Neotropics, (B) Africa (includes Madagascar), (C) Southeast Asia, and (D) South Asia.

# RESULTS

## Commodity-driven deforestation

Data from the Global Forest Watch database on forest cover loss (>30% canopy cover) and commodity-driven deforestation for the period 2001–2017 show that both forest loss and commodity-driven deforestation were widespread in countries in the four primate-range regions (Table 1; Figs. 2A–2D). The Neotropics was the region with most forest loss (83.5 million ha) followed by Southeast Asia with 54.3 million ha (Table 1). Africa ranked third with 38.5 million ha, and South Asia fourth with 1.9 million ha (Fig. 2; Table 1).

Total estimated forest loss for these regions over this 15-year period was 178.8 million ha, with an annual rate of forest cover loss of 10.52 million ha/year (Table 1). The average percent of commodity-driven permanent deforestation for the period 2001–2015 was highest in Southeast Asia (47%) and lowest in Africa (7%) (Fig. 2; Table 1).

Globally, the primate-range countries with the highest percentage of commodity-driven deforestation relative to total forest loss for the period 2001–2015 were Malaysia (91% of 7.3 million ha), Paraguay (90% of 5.5 million ha), Indonesia (89% of 24.4 million ha), Argentina (82% of 5.2 million ha), Bolivia (70% of 4.5 million ha), and Brazil (66% of 51 million ha) (Fig. 2; Table 1; Fig. S1). Many of these countries have high primate diversity. In the Neotropics, Brazil (the primate-richest country with 116 species) stands out among the top five commodity producing countries in the region due to the disproportionate degree of forest loss during 2001–2017 (Fig. 2A). In Africa, the DRC, with 49 primate species, experienced the greatest amount of deforestation, mainly due to shifting agriculture and the extraction of forestry products (Fig. 2B). The DRC was followed by Madagascar, Mozambique, Angola, Ivory Coast, and Tanzania (Fig. 2B). While shifting agriculture was a dominant cause of deforestation in Africa, several countries in this region were heavily involved in the production of export commodities (e.g., Madagascar, Ivory Coast, Liberia, Nigeria Uganda, Kenya, Equatorial Guinea, and Rwanda among others) (Fig. 2B; Table S1). During the period 2001–2017, Madagascar deforested more land (3.7 million ha) than all mainland African countries except for the DRC. A total of 27% of forest loss in Madagascar was due to commodity deforestation (Fig. 2B). This is consistent with a recent study based on remote sensing indicating that between 2001 and 2015, 27% of global forest loss (>30% canopy cover) worldwide was attributed to deforestation through permanent land use change for commodity production. The remaining loss was attributed to forestry (26% was due to natural forests being cut down and trees being planted for wood-based products), shifting agriculture (24%), and wildfires (23%) (*Curtis et al., 2018*). Over the last three to four decades, industrial agribusiness and industrial logging have been producing wood products for global rather than local markets (*Food and Agriculture Organization of the United Nations (FAO), 2009*), with the result that commodity-driven agriculture and logging are, currently, the most significant human-induced drivers of forest loss and forest degradation worldwide (*Hosonuma et al., 2012*). Notwithstanding corporate pledges, the rate of commodity-driven forest loss has not lessened and, in order to end deforestation, companies will need to eliminate five million hectares of converted forest from supply chains each year (*Curtis et al., 2018*).

## International commodities trade

### Growth of commodity exports by countries in primate-range regions

Trade data show that, between 2000 and 2016, agricultural and nonagricultural commodity exports increased by a factor of two to four for primate-range countries, suggesting a global trend in the growth of market demands, higher production of commodities, and an expanding land use footprint by importing nations (Fig. 3A). These data also indicate that, in both 2000 and 2016, a significant share of these imports was purchased by countries

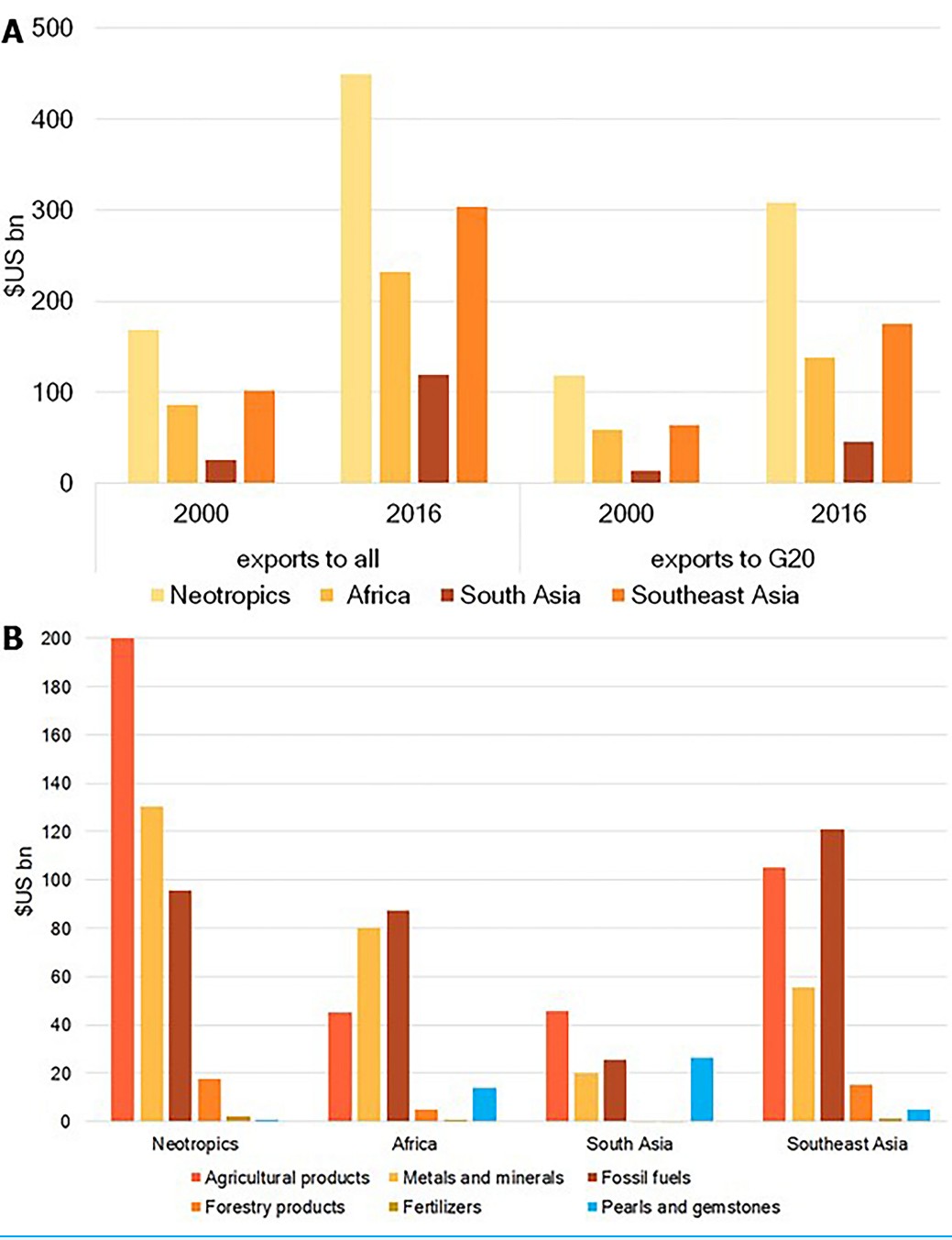

**Figure 3 Commodity exports.** (A) Growth of commodity exports between 2000 and 2016 by countries in primate-range regions. Exports refer to all trading patterns (left side of figure) and to countries that comprise the G-20 group (right side of figure). Source: resource trade.Earth of Chatham House. (B) Commodities exported in 2016 by countries in each primate range region. Beef is considered an agricultural commodity as its large-scale production requires the conversion of large areas of forest into pasture.

in the G-20 group (Fig. 3A). In 2016, for example, primate-range regions exported US$ 1.1 trillion in natural resource commodities. The G-20 group took 60% of commodity exports by countries in the four primate-range regions. This is consistent with the fact that, in

2016, demand from G-20 countries accounted for over half of all of the world's resource imports (https://resourcetrade.earth/) and emphasizes the disproportionate land use footprint of a few consumer nations. A list of countries in the G-20 group can be found in Table S2.

In 2016, countries in all primate regions were commodity exporters, but the relative importance of particular commodities varied from region to region. In the Neotropics, the commodity groups most commonly exported were agricultural products, metals and minerals, and fossil fuels (Fig. 3B). In Africa, metals, minerals, and fossil fuels were the predominant exports. In both South Asia and Southeast Asia the principal exports were fossil fuels and agricultural products. Forestry products were a commonly exported commodity in Southeast Asia (Fig. 3B). Clearly, global market demands for commodities are an important source of income for the economies of primate-range countries. This has come at the cost of the permanent conversion of natural habitats into anthropogenic landscapes, causing the loss of biodiversity and significant declines in primate populations (Fig. 2) (*Chaudhary & Brooks, 2017*; *Chaudhary & Mooers, 2018*; *Estrada et al., 2018*).

### Trade in commodities leading to permanent deforestation

#### Regional patterns

Below, we profile the international trade of soybeans (*Glysine max*), palm oil (*Elaeis guineensis*), and natural rubber (*Hevea brasiliensis*) that results in permanent deforestation. We also examine beef production, a major driver of the conversion of forests to pasture. Data, from resourcetrade.Earth, on the international trade in these four commodities, indicate exponential growth (US bn) between 2000 and 2016, and this has differentially affected permanent deforestation across primate range regions (Fig. 4). For example, in both 2000 and 2016, soybeans dominated exports in the Neotropics. This region was also the principal exporter of beef during that period (Fig. 4). Southeast Asia was the leading exporter of palm oil and rubber in both 2000 and 2016. The other primate range regions contributed smaller volumes to the trade in these commodities (Fig. 4).

#### Expansion of land area for production of agricultural commodities between 1960 and 2016

Data from FAOSTATS show that the production of soybeans in the Neotropics and Southeast Asia has steadily increased since 1960. In 2016, soybean production involved almost 74 million ha of land in primate range countries, of which some 80% was in the Neotropics, particularly Brazil and Argentina (Fig. 5A). In 2016, some 15 million ha of land in Southeast Asia were devoted to palm oil production (about 20 million ha worldwide). The conversion of forested land for the production of palm oil in Southeast Asia has increased steadily since 1970 (Fig. 5B). Overall, soybeans and oil palm have become increasingly important not only as food products for humans and domestic animals, but also for biofuel production (*Koh & Ghazoul, 2008*). In 1960, 4 million ha worldwide were devoted to the cultivation of natural rubber. By 2016, this had increased to 11.4 million ha, with Southeast Asia accounting for about 78% of the total (Fig. 5C).

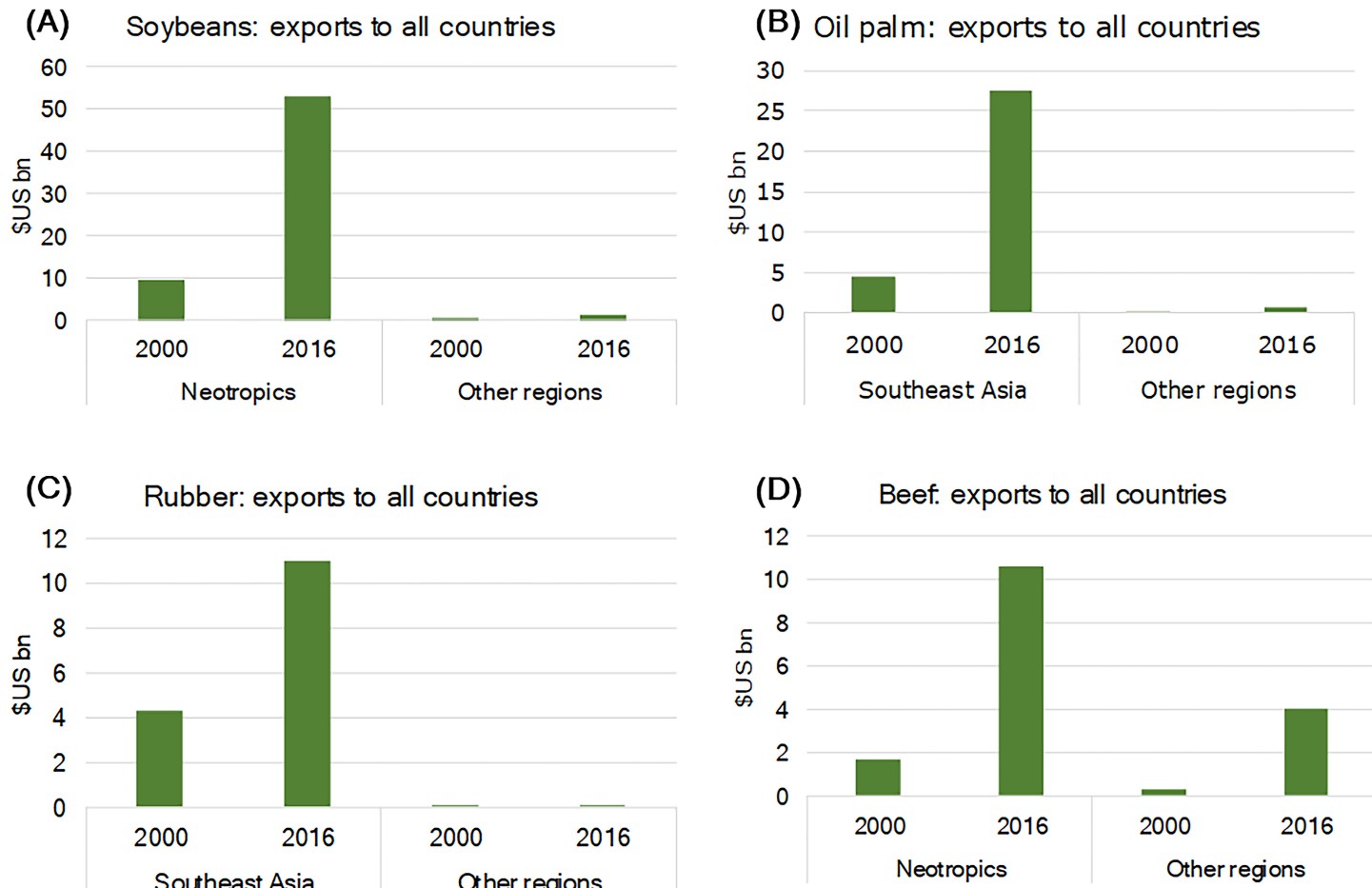

**Figure 4** Growth in the export of (A) soybeans, (B) oil palm, (C) natural rubber, and (D) beef that led to permanent deforestation in primate-range regions between 2000 and 2016. Other regions in (A) and (D) refer to Africa, South Asia and Southeast Asia and in (B) and (C) to the Neotropics, Africa, and South Asia.

## Trade flows in 2016 of agricultural commodities resulting in permanent forest loss

*Soybeans*

In 2016, a total of US$48.72 bn in soybeans was exported by primate-range countries (this involved almost 74 million ha of land), of which 92% were exports by Brazil and Argentina, two countries in the Neotropics (Fig. 6A). Brazil, with 116 primate species, was the second largest producer of soybeans in the world (the US was the top producer, *USDA, 2018*), and contributed approximately 80% of the soybean crop exported by primate-range countries (https://globalforestatlas.yale.edu/amazon/land-use/soy). In Brazil, 30% of deforestation between 2000 and 2010 was driven by global demand for soybeans and beef exports (*Karstensen, Peters & Andrew, 2013*). While direct deforestation for soybean production in the Amazon has remained low, it has become progressively common in the *Cerrado* region of Brazil (*Beckman et al., 2017*; *Gibbs et al., 2015*; *Zalles et al., 2019*), where 28 primate species are found (IUCN Red List 2019-1, 2019). Other primate-habitat

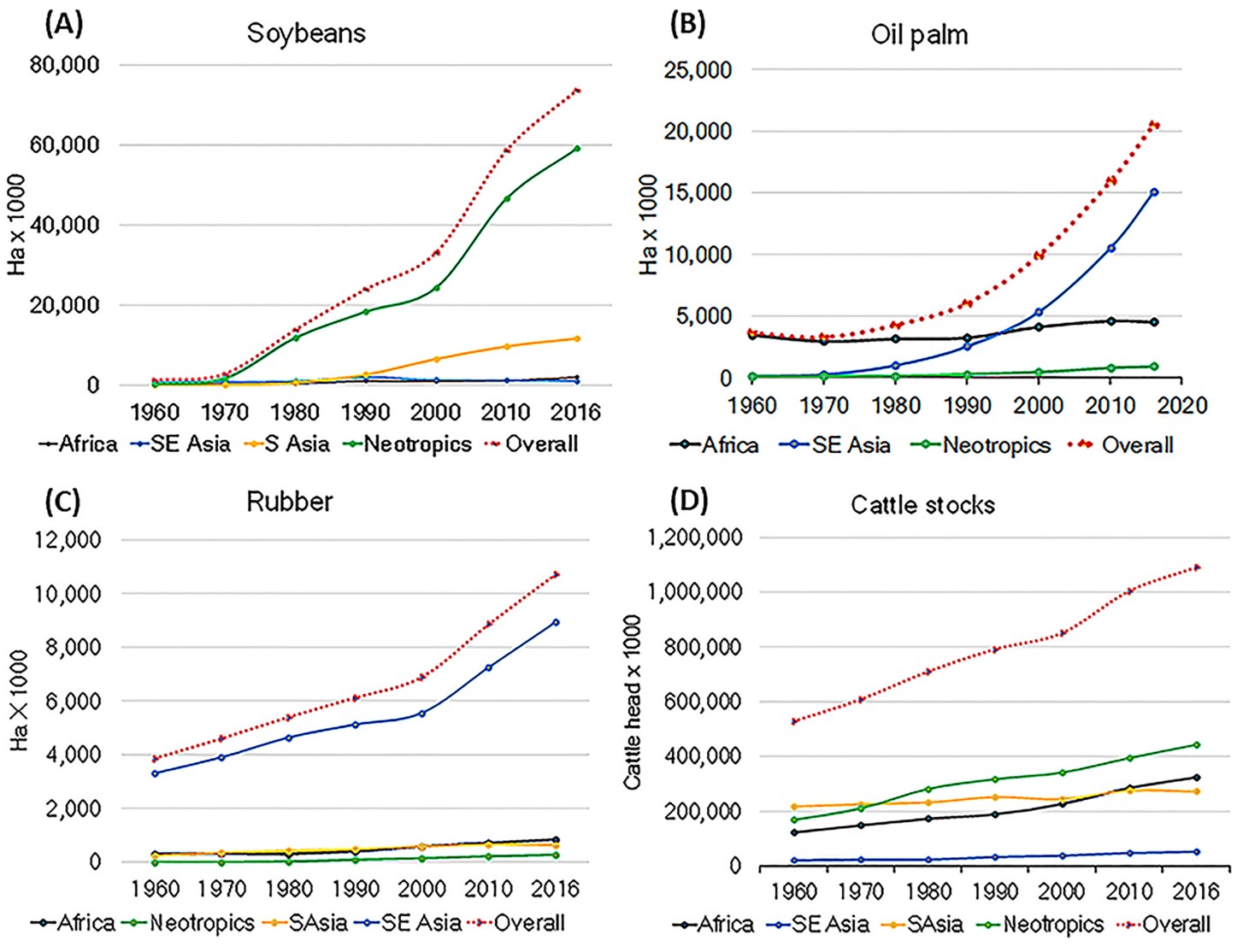

**Figure 5 Increase in the area of land dedicated to the production of (A) soybeans, (B) oil palm, (C) natural rubber, and (D) cattle stocks.** These increases led to permanent deforestation in primate-range regions between 1960 and 2016.

countries that are significant exporters of soybeans are Argentina, Bolivia, and Paraguay. Most deforestation in Bolivia has been in the Amazon basin where expansion of soybean agriculture has intensified and where 24 primate species occur (*Muller et al., 2012*; *Zalles et al., 2019*). Other countries in Africa, South Asia, and Southeast Asia contributed lesser amounts to the international trade in soybeans (Fig. 6A). China led the import market in 2016 (Fig. 6B), followed by India, the Netherlands, and Vietnam. Much smaller amounts of soybeans were purchased by countries in South Asia, Southeast Asia, and Africa (Fig. 6B). In Brazil ca 70% of soybean exports in 2016 was turned into soymeal used as livestock feed (*Delgado, 2005*; *Fuchs et al., 2019*; *Mielke & Mielke, 2018*) and soy oil (a derivative of soymeal production), which is used by the food industry to produce soy sauce, cooking oil, miso, soy milk, soy curd, tempeh, and tofu products. It is also used in

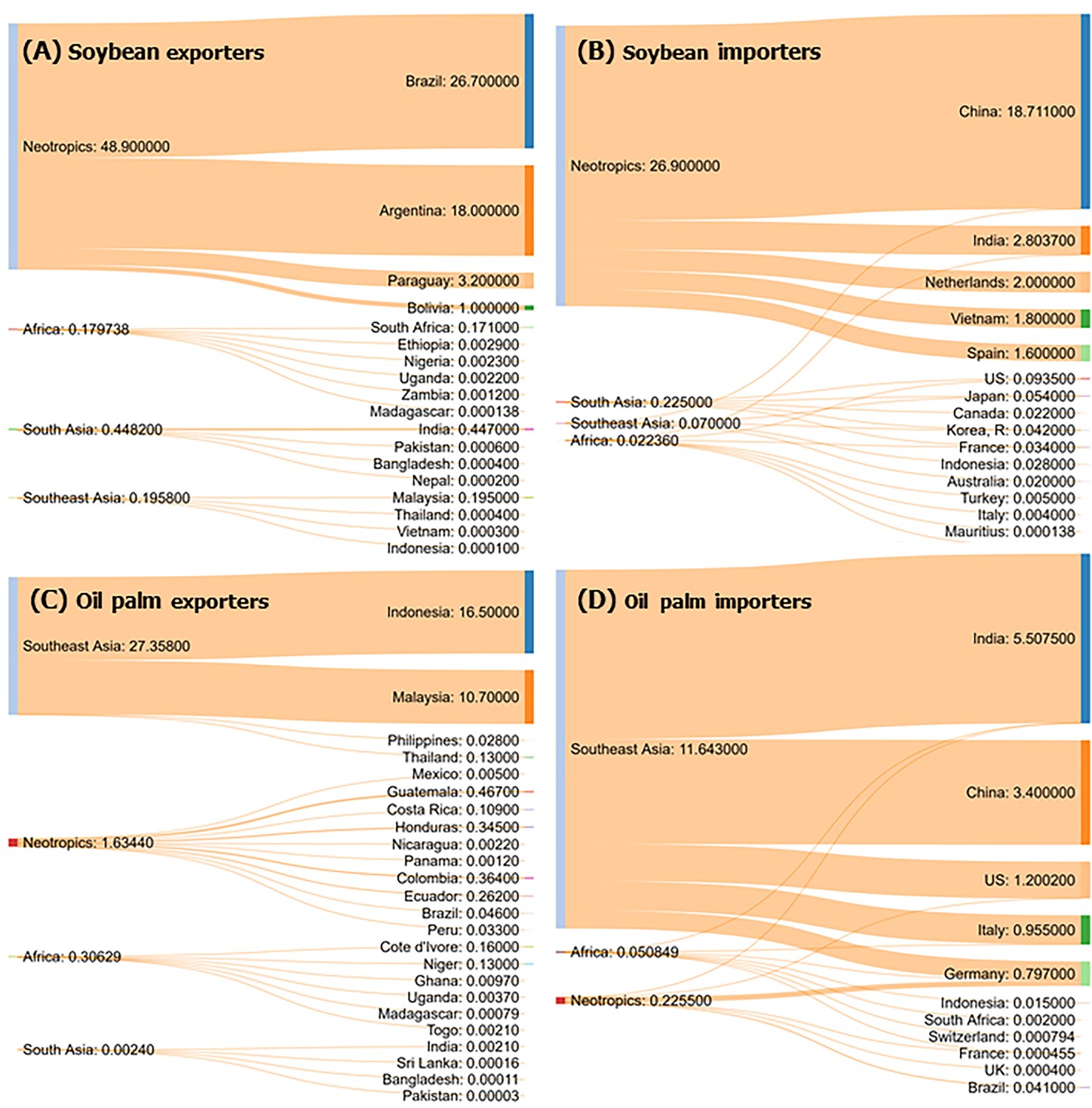

**Figure 6 Trade flow diagrams for the top exporter and importer countries of four commodities that resulted in permanent deforestation in each primate-range region in 2016.** Soybeans (A and B), oil palm (C and D). Numbers on the flow diagrams indicate the amount in US$ bn of trade for each country. For exports and imports, the trade flows show on the left the accumulated US$ bn for each region and for the countries involved. The width of the connecting flows is proportional to the value exported or imported.

the production of detergents, cosmetics, and various industrial chemicals (*Casson, 2003*; *USDA-FAS, 2018*).

*Oil palm*

The total accumulated value of exports of palm oil by primate-range nations in 2016 was US$15.42 bn, with 95% of the production from Indonesia and Malaysia (Fig. 6C). Clear-cutting for wood and to establish oil palm plantations are common patterns in these two countries (*Henders, Persson & Kastner, 2015*). With 56 species, Indonesia is the primate richest country in Southeast Asia. Malaysia has 20 primate species (*Estrada et al., 2018*). Oil palm imports from Southeast Asia were purchased, in 2016, principally by India, China, the US, Italy, and Germany (Fig. 6D). Primate-range countries in the other regions contributed smaller amounts to the palm oil trade. Data from FAOSTATS show that cultivation of oil palm has expanded greatly in Southeast Asia from 5.5 million ha in 2000 to 15 million ha in 2016 (Fig. 5B). Growing global demand for oil palm products is a major cause of rapid decline in the remaining populations of the Critically Endangered Sumatran and Bornean orangutans (*Pongo abelii* and *Pongo pygmaeus*, respectively) and a serious risk for the Endangered or Critically Endangered chimpanzees, bonobos, and gorillas in Africa, because a large proportion of extant populations are found outside protected areas (*Grasp & IUCN, 2018*; *Lanjouw, Rainer & White, 2015*; *Linder, 2013*; *Vijay et al., 2016*; *Wich et al., 2014*). Projections of land-cover change indicate that the Bornean orangutan, for example, is expected to lose 15–30% of its remaining habitat by 2080, mainly due to deforestation and oil palm plantations (*Nantha & Tisdell, 2009*; *Struebig et al., 2015*; *Vijay et al., 2016*; *Wich et al., 2014*). Moreover, given the growing global demand for palm oil, which is expected to convert some 400 million ha of African forest to monoculture by the year 2050, population decline and habitat loss is projected to threaten over 40 species of African primates (*Strona et al., 2018*). The oil palm, native to West Africa, is now cultivated in large-scale plantations across the tropics. It is used in several commercial products including cooking oil, soap, cosmetics, and margarine (*USDA-FAS, 2018*).

*Natural rubber*

The total accumulated value of exports of natural rubber by top exporter primate-range countries in 2016 was US$11.1 bn and the most important primate-range region for supplying natural rubber to global markets was Southeast Asia, with Thailand and Indonesia contributing to 76% of the total value of exports (Fig. 7A). Major importing countries were India, China, and the US (Fig. 7F). Exports of rubber from Southeast Asia went from 4.3 US$ bn in 2000 to almost 11 US$ bn in 2016, a reflection of the dramatic expansion of rubber plantations over the past several decades (Fig. 5C). In 1960, four million ha worldwide were devoted to rubber cultivation and, by 2016, the area of land converted to rubber plantations had reached 11.4 billion ha, with Southeast Asia accounting for approximately 78% (Figs. 5C and 8). Deforestation due to the establishment of rubber plantations in South Asia and China has severely affected populations of threatened primates such as the Vulnerable Bengal slow loris (*Nycticebus bengalensis*),

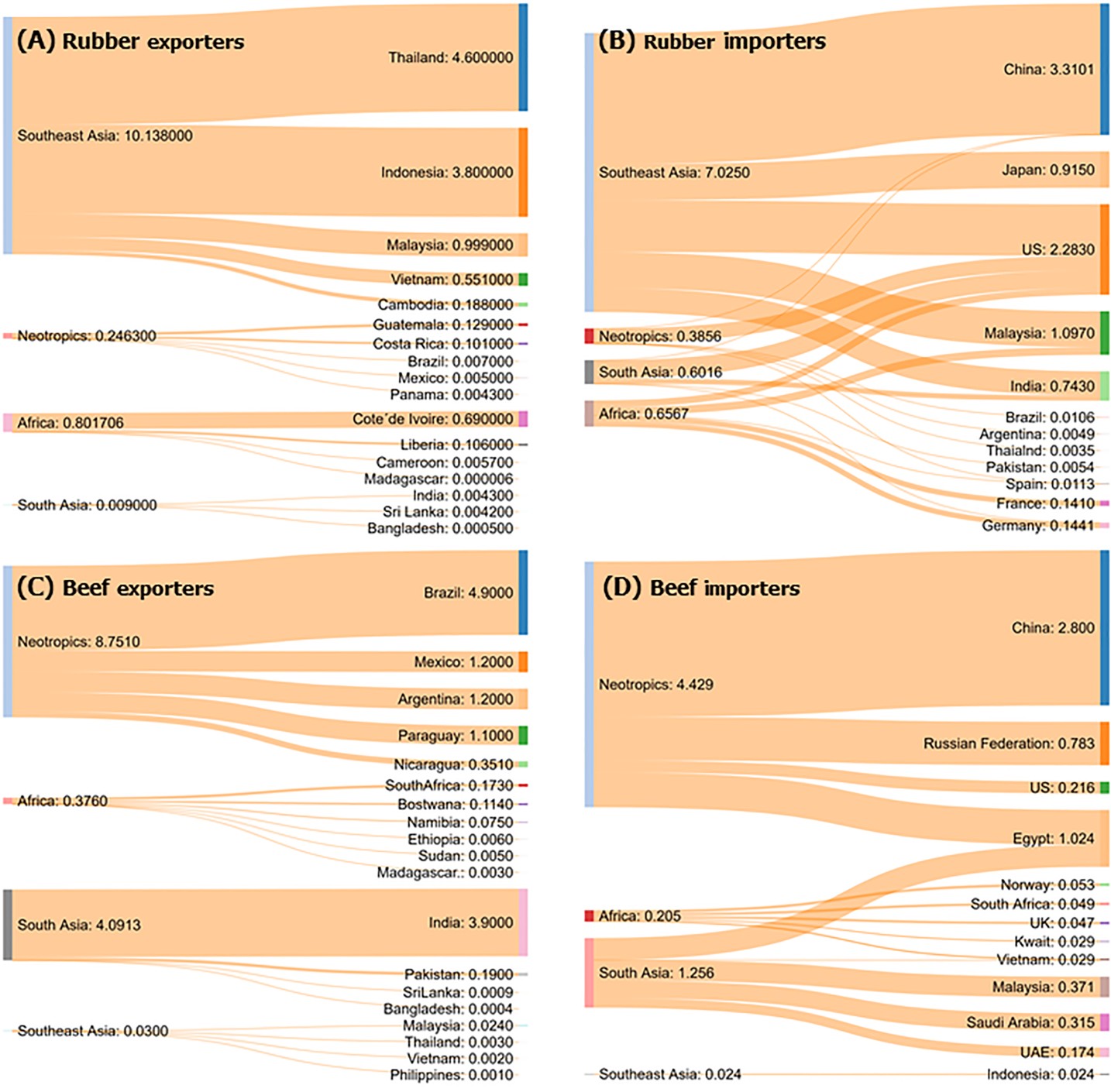

**Figure 7 Trade flow diagrams for the top exporter and importer countries of four commodities that resulted in permanent deforestation in each primate-range region in 2016.** Natural rubber (A and B) and beef (C and D). Numbers on the flow diagrams indicate the amount in US$ bn of trade for each country. For exports and imports, the trade flows show on the left the accumulated US$ bn for each region and for the countries involved. The width of the connecting flows is proportional to the value exported or imported.

the Endangered western hoolock gibbon (*Hoolock hoolock*), the Endangered Phayre's langur (*Trachypithecus phayrei*), the Critically Endangered northern white-cheeked crested gibbon (*Nomascus leucogenys*), and the Critically Endangered Hainan gibbon

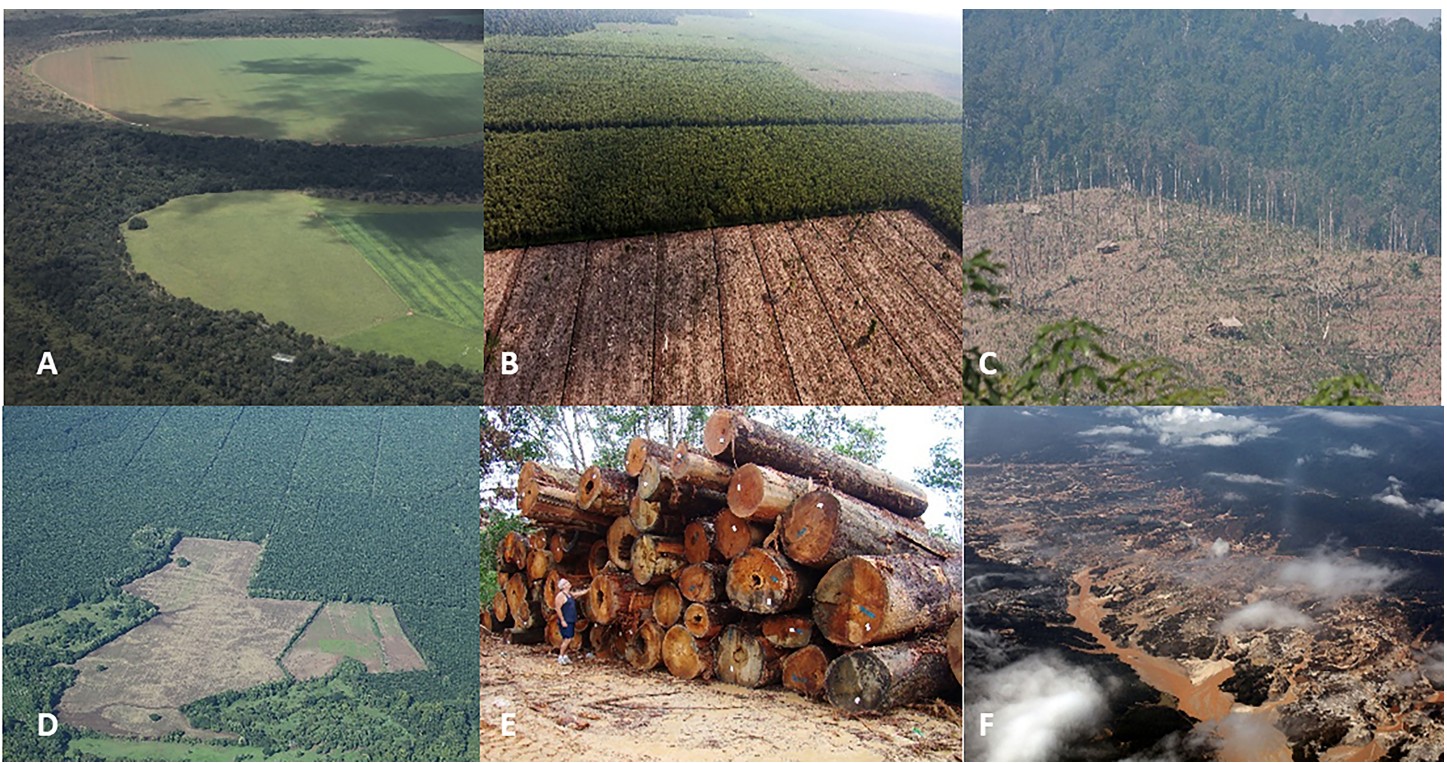

**Figure 8 Photos of selected forest-risk commodities in primate range regions.** Photo credits include the following: (A) forest converted to soy fields in Brazil (photo credit: R. Butler), (B) pulp and paper plantation in Indonesia (photo credit: R. Butler), (C) deforestation for natural rubber production in Laos (photo credit: R. Butler), (D) oil palm plantation in Costa Rica (photo credit: R. Butler), (E) industrial logging in Malaysia (photo credit: W. F. Laurance. (F) Gold mining in Peru (photo credit: R. Butler).

(*Nomascus hainanus*) (<30 individuals left; *Fan, Fei & Luo, 2014*; *Mazumder, 2014*). Even protected areas are at risk due to the expansion of rubber plantations. Between 2005 and 2010, over 2,500 km$^2$ of natural tree cover and 610 km$^2$ of protected areas were converted to rubber plantations across Southeast Asia (*Ahrends et al., 2015*). Growing global market demands for natural rubber are driving both industrial-scale and smallholder monocultures of this commodity, with >2 million ha established in the last two decades, mostly in Southeast Asia and southwest China, including the primate-rich province of Yunnan (*Li et al., 2018*). It is estimated that an additional eight million ha of rubber plantations will be required to meet world demand by 2024 (*Warren-Thomas, Dolman & Edwards, 2015*). Global demand for natural rubber also has increased rapidly in the past decades, with 70% of global consumption used for tires. Rubber is the most rapidly expanding tree crop in mainland Southeast Asia (*Ahrends et al., 2015*; *Warren-Thomas, Dolman & Edwards, 2015*).

*Beef*
The total accumulated value of exports of beef by top exporter primate range nations in 2016 was US$13.2 bn. The most important primate-range region supplying beef to global markets in 2016 was the Neotropics. In both the Neotropics and South Asia, beef exports increased significantly between 2000 and 2016 (Fig. 4D). Trade flows show that
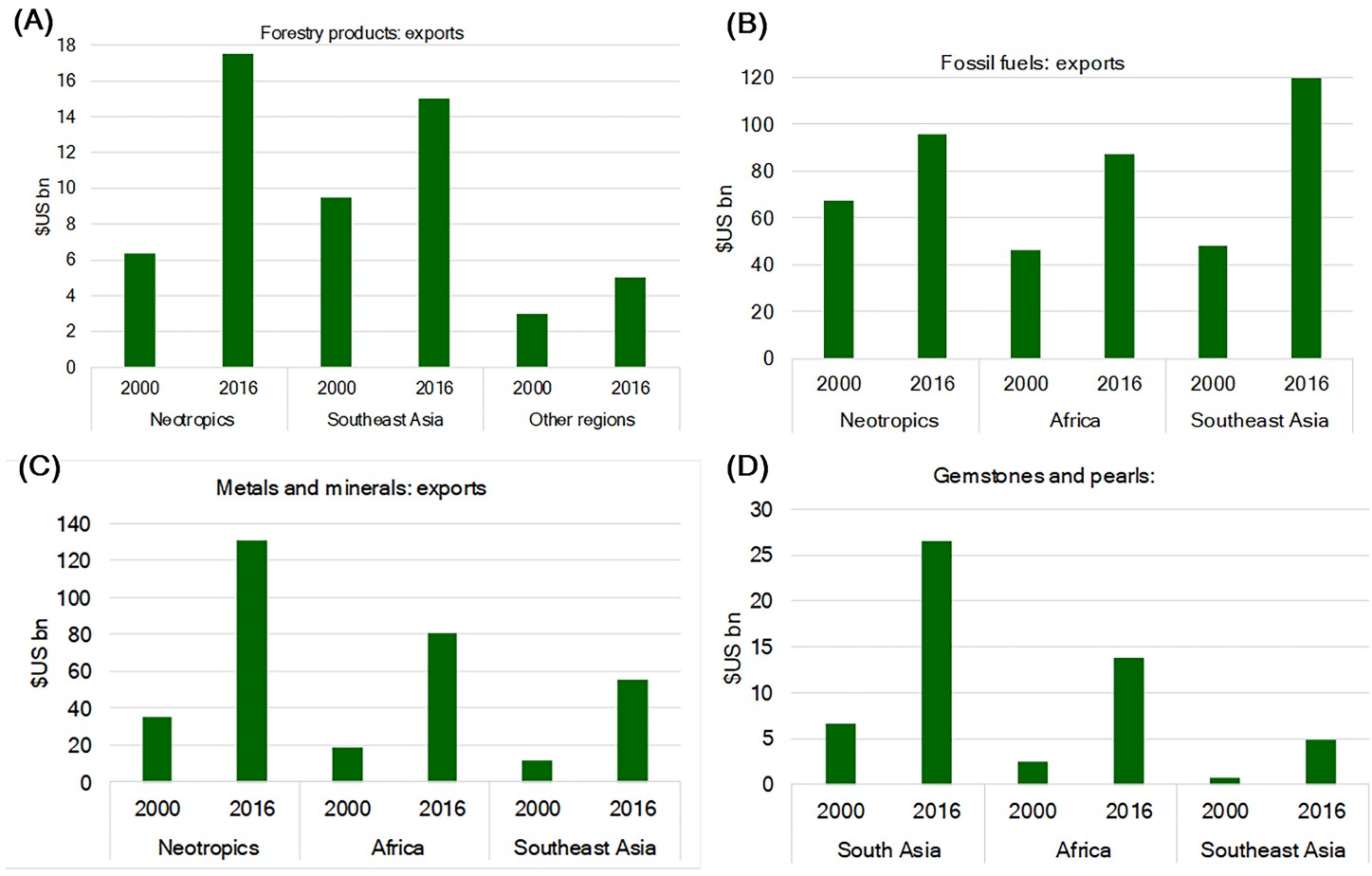

**Figure 9 Growth in the export value of (A) forestry products, (B) fossil fuels, (C) metals and minerals, and (D) gemstones and pearls in primate-range regions between 2000 and 2016.**

beef exports in the Neotropics were dominated by Brazil, Mexico, Argentina, and Paraguay, accounting for 63% of beef exports from this region (Fig. 7C). In 2016, China, the Russian Federation, and Egypt were the top importers of beef from the Neotropics, suggesting that these countries drive the destruction and conversion of large areas of forested land in this region (Fig. 7D). In South Asia, India was the major beef exporter, accounting for 95% of exports (Fig. 7C). Imports of South Asian beef went principally to Egypt, Malaysia, Saudi Arabia, and the UAE (Fig. 7D). As a result of internal and global demands for beef, cattle stocks have been growing rapidly since 1980 in both the Neotropics and South Asia, and at a slower rate in Africa and Southeast Asia. However, collectively in these four primate regions, cattle stocks have grown from 527.7 million head in 1960 to 1 billion head in 2016 (Fig. 5D), with concomitant increases in pasture land and extensive use of rangelands placing many primate populations at risk (*Estrada et al., 2017*).

### Trade in nonagricultural commodities leading to forest degradation

The international trade in forestry products (e.g., lumber and charcoal), land-based fossil fuels, metals and minerals, and gemstones from primate-range regions is a major driver of forest degradation and loss (https://comtrade.un.org). Trade data indicate that the export

of these commodities grew substantially between 2000 and 2016 (Figs. 9A–9D). The Neotropics and Southeast Asia were major exporters of both forestry products and fossil fuels. Major exporting regions of metals and minerals were the Neotropics and Africa, while major exporters of gemstones and pearls were South Asia and Africa (Figs. 9A–9D).

### Trade flows in 2016 of nonagricultural commodities resulting in forest degradation

*Forestry products*

The total accumulated value of exported forestry products by the top exporting primate-range nations in 2016 was US$21 bn. Trade flow data show that while all four of the primate-range regions exported forestry products, the Neotropics and Southeast Asia were the greatest exporters (Fig. 10A). In the Neotropics, the top exporter of forestry products in 2016 was Brazil, and the top importer nations were China and the US, followed by Italy, Germany, and Japan (Fig. 10B). Top exporters of forestry products in Southeast Asia were Indonesia and Malaysia, followed by Thailand and Vietnam (Fig. 10A). China, Japan, and South Korea were the top importers of forestry products from Southeast Asia (Fig. 10B). Major importers of forestry products from Africa were China, India, Belgium, and France (Fig. 10B). In addition, the total accumulated export of wood from the Congo Basin countries—which have some 60 primate species—to China doubled between 2001 and 2015, with a concomitant increase in the loss of tree cover (*Fuller et al., 2018*). Global demand for tropical forestry products (e.g., timber) has increased over the past several decades. This has led to an expansion of logging activities creating a potent economic incentive for road building in otherwise forested and hard to access areas in primate-range regions (*Malhi et al., 2014*). Data from FAOSTAS show that harvesting industrial nonconiferous roundwood increased by an order of magnitude between 1960 and 2016 (from 34 million $m^3$ in 1960 to 286 million $m^3$ in 2000 to 389 million $m^3$ in 2016, Fig. S2), in response to the ever-increasing worldwide demand for tropical timber. Complex environmental and economic drivers surround the trade of tropical timber and deforestation. For example, US demand for Chinese-made furniture positively correlated with Chinese timber imports from the Congo Basin, suggesting that the US demand for furniture motivates the harvest of timber by Chinese commercial businesses (*Fuller et al., 2018*). Although some primate species—orangutans (*Pongo*), for example—can survive temporarily in logged forests, they and other primate species such as chimpanzees (*Pan troglodytes troglodytes*) and gorillas (*Gorilla gorilla gorilla*) are adversely affected, and their long-term persistence in these degraded habitats is unlikely (*Meijaard et al., 2012*; *Voigt et al., 2018*; *Morgan et al., 2018*). Logging reduces canopy cover and forest undergrowth, and consequently humidity. The drying of the subcanopy and undergrowth increases tree mortality and the likelihood of ground fires, affecting the regeneration of the large tree species that provide food, resting sites, and refuge for primates (*Alisjahbana & Busch, 2017*; *Lewis, Edwards & Galbraith, 2015*; *Peres, 1999*, *2001*; *World Bank, 2018*). Moreover, the influx of hunters, miners, and settlers has led to the pollution of streams and an increase in bushmeat hunting (*Remis & Robinson, 2012*).

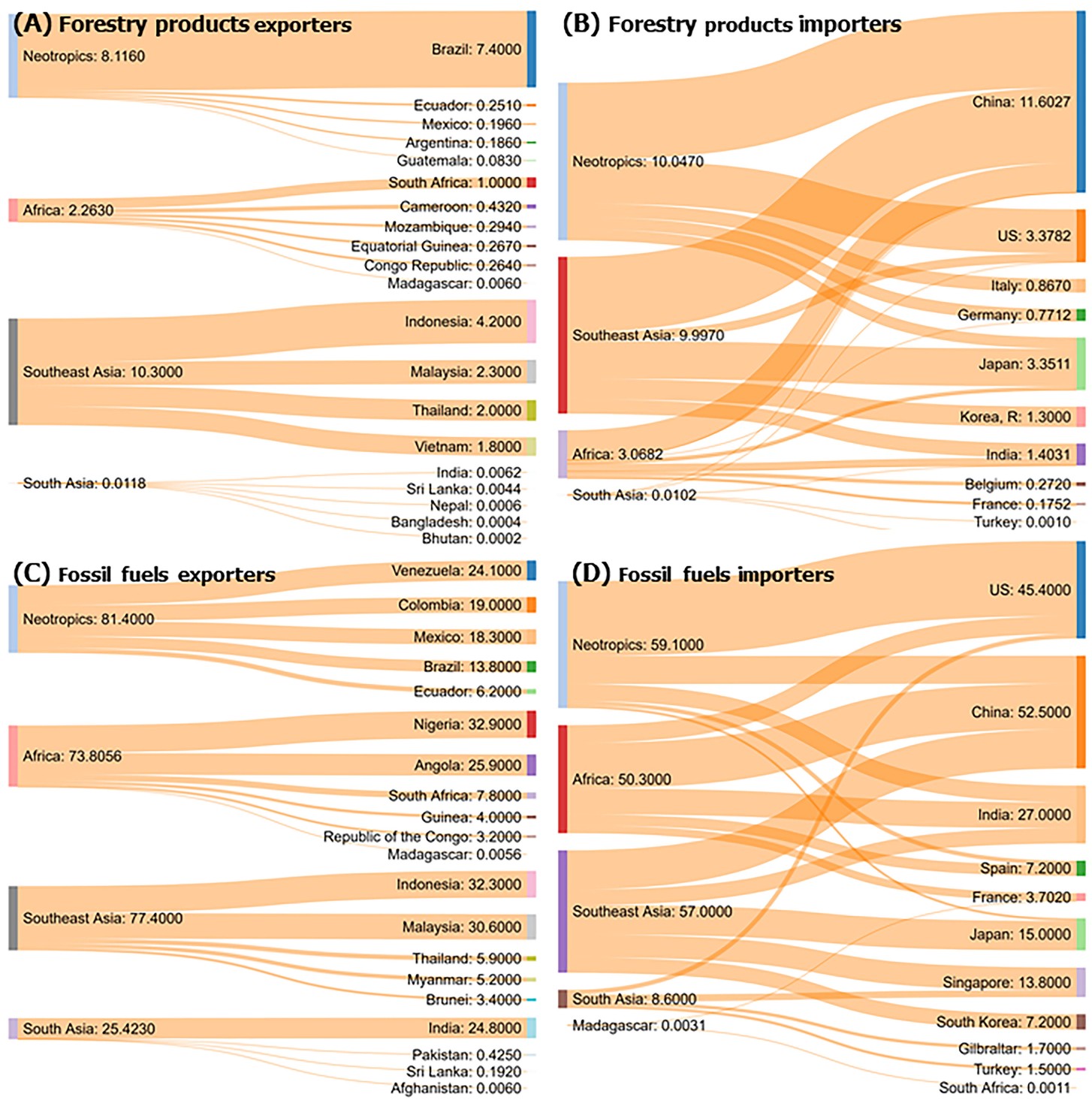

**Figure 10** Trade flow diagrams of four commodities for the top exporter and importer countries and primate range regions that resulted in forest degradation in 2016. Forestry products (A and B), fossil fuels (C and D). We show only the top exporting and importing countries in each region. Numbers on the flow diagrams indicate for each country, the amount in US\$ bn involved in trade. For exports and imports the trade flows also show on the left the accumulated US\$ bn for each region and for the countries involved. The width of the connecting flows is proportional to the value exported or imported.

*Fossil fuels (coal, gas, oil)*

The total accumulated value of exports of fossil fuels by the top exporting primate-range nations in 2016 was US$258 bn (Fig. 10). Fossil fuel exports increased markedly in the Neotropics, mainland Africa, and Southeast Asia between 2000 and 2016 (Figs. 9A–9C). In the Neotropics, Venezuela, Colombia, and Mexico (Fig. 10C) were the leading exporting nations. The leading importer nations were the US, China, and India (Fig. 10D).

In mainland Africa, exports of fossil fuels in 2016 were greatest in Nigeria and Angola, and major importers were China, India, and the US (Fig. 10C). In Southeast Asia, Indonesia, and Malaysia were the main exporters of fossil fuels, with China, Japan, Singapore, and South Korea being the major importers (Fig. 10D). In South Asia, India was the largest producer of fossil fuels and the major importers were Singapore and the US (Figs. 10C and 10D). Global demand for oil and natural gas is expected to grow between 30% and 53% by 2035, and primate-rich areas such as the Amazon, Malaysia, and Borneo will be severely affected (*Finer et al., 2015*). Oil and gas concessions in the western Brazilian Amazon and in forested areas of Colombia, Ecuador, Perú, and Bolivia, already cover about 733,414 km$^2$ (*Finer et al., 2015*). At the time of writing, 327 oil or gas blocks, covering some 108 million ha, are projected or are being explored in the Amazon Basin (*Bebbington et al., 2018*). In many cases, these potential energy fields have considerable overlap with protected areas, and currently exploited oil and gas infrastructure tends to be found on land where biodiversity, species richness, and range rarity are high (*Harfoot et al., 2017*). This exacerbates the negative effects that fossil fuel extraction has on primate populations. Human societies are largely dependent on fossil fuels (coal, oil, and natural gas), which has contributed to increased carbon in the atmosphere and to climate change. The land-based extraction of fossil fuels, similar to the negative impact of the harvesting of forest products, results in decreased biodiversity and increased accessibility for settlements, logging, hunting, and agriculture, all resulting in primate habitat loss and pollution (*Harfoot et al., 2017*; *Finer & Orta-Martínez, 2010*).

*Metals and minerals*

The total accumulated value of exports of metals and minerals by the top exporting primate-range nations in 2016 was US$204 bn (Figs. 11A–11D). The Neotropics was the leading primate region exporting metals and minerals (US$85.6 bn), followed by Africa (US$54.5 bn), Southeast Asia (US$35.0 bn), and South Asia (US$19.8 bn) (Fig. 11A). Of the countries in the Neotropics, Brazil, Peru, and Mexico were the major exporters of metals and minerals. In Africa, South Africa dominated metal and mineral exports (Fig. 11A). Major exporters in Southeast Asia were Indonesia, Thailand, and Malaysia (Fig. 11A). Pakistan, India, and Afghanistan were major exporters in South Asia (Fig. 11A). In 2016, China was the primary importer of metals and minerals from the Neotropics, Africa, Southeast Asia, and South Asia. Other leading importers were the US, Switzerland, Japan, South Korea, and India (Fig. 11B). The mining of metals and minerals is a persistent threat to primates and their habitats. Mining contributes to habitat destruction, fragmentation, deforestation, and the poisoning and pollution of soil and ground water (*Alvarez-Berríos & Mitchell Aide, 2015*; *Garcia et al., 2017*). In Brazil, 126 mining dams are

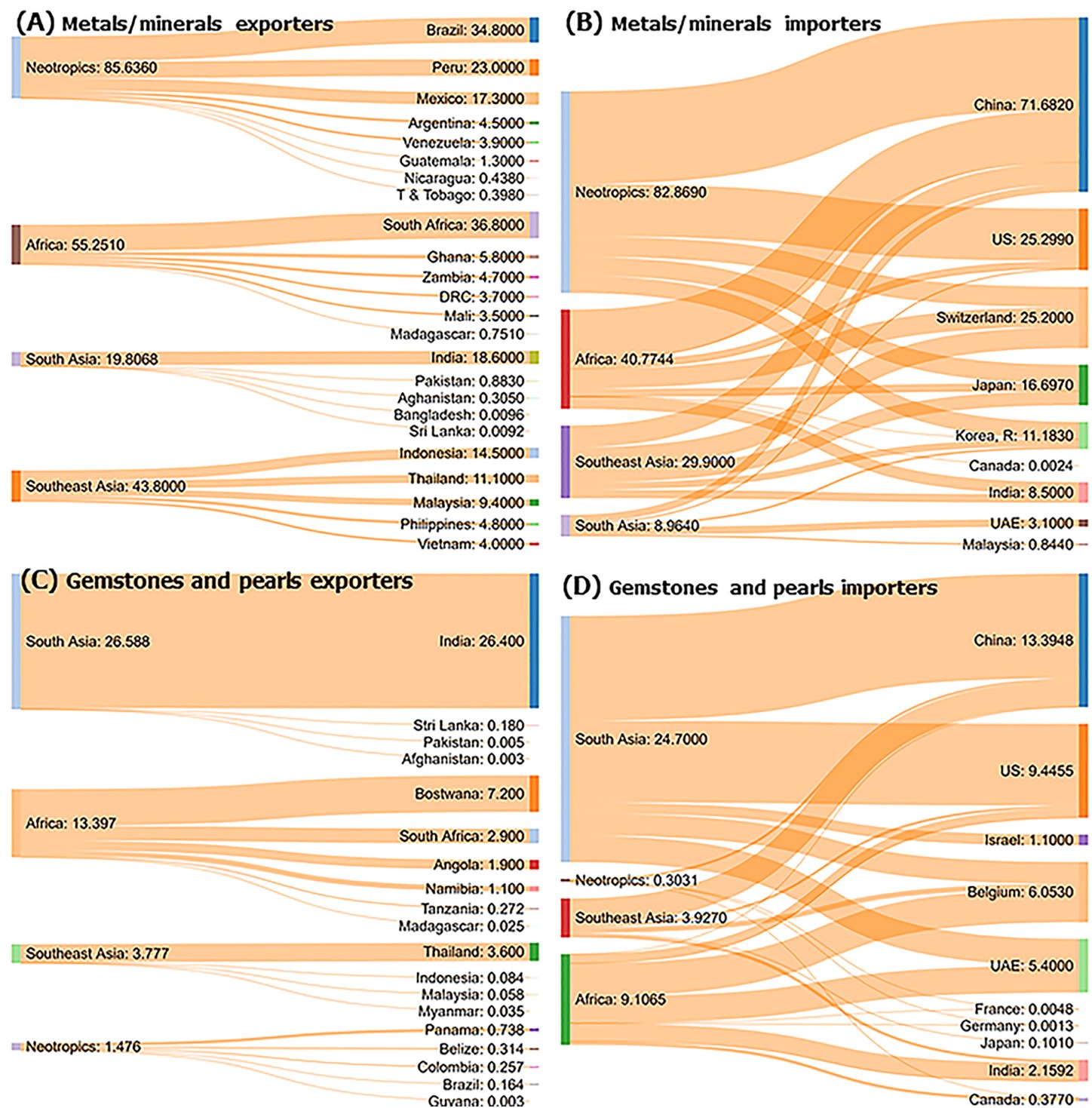

**Figure 11 Trade flow diagrams of four commodities for the top exporter and importer countries in primate range regions that resulted in forest degradation in 2016.** Metals and minerals (A and B) and gemstones and pearls (C and D). We show only the top exporting and importing countries in each region. Numbers on the flow diagrams indicate for each country, the amount in US$ bn involved in trade. For exports and imports the trade flows also show on the left the accumulated US$ bn for each region and for the countries involved. The width of the connecting flows is proportional to the value exported or imported.

currently at risk of failing. In one case, dam failure poisoned hundreds of kilometers of the Rio Doce, from the upper reaches to the sea, with a toxic mud of iron-ore tailings (*Garcia et al., 2017*). Added effects are extensive tree mortality on the borders of both small and large mines, the establishment of camps and frontier towns, the construction of access roads, rail, and trails, the construction of mining dams, and hydro-power/waterway developments (*Alvarez-Berríos & Mitchell Aide, 2015*; *Asner et al., 2013*; *Global Forest Watch, 2018*). Mining stimulates human migration, resulting in illegal logging and the colonization of forested areas, with the consequent increase in bushmeat hunting and illegal primate trade (*Alamgir et al., 2017*).

In eastern DRC, unprotected areas of high animal and plant biodiversity overlap with areas that are rich in minerals (*Edwards et al., 2014*). Increased global demand for surface deposits of conflict minerals such as tantalum, coltan, and gold has resulted in the expansion of illegal mining camps in several national parks in the DRC. Widespread bushmeat hunting in these areas has devastated populations of Grauer's gorillas, eastern chimpanzees, and other primate species (*Plumptre et al., 2015*; *Spira et al., 2017*). Illegal gold mining in Madagascar by itinerant miners has impacted many forests inhabited by lemurs (*Duffy, 2007*). In the Philippines, mining of gold, nickel, and copper on Dingaan Island has endangered the survival of tarsiers (*Carlito syrichta*) (*Brown et al., 2014*). In Ghana, mining-induced hunting and logging have caused the decline of primates such as the Dwarf galago (*Galagoides demidoff*), Bosman's potto (*Perodicticus potto*), and the Mona monkey (*Cercopithecus mona*) in forest reserves adjoining mine sites (*Erasmus et al., 2018*). Gold mining in Indonesia is a major threat to the Endangered proboscis monkey (*Nasalis larvatus*) (*Meijaard & Nijman, 2000*) and to Bornean orangutans and the Endangered Bornean gibbons (*Hylobates muelleri*) (*Garcia et al., 2017*; *Lanjouw, 2014*). Pressure to expand mining as a result of global market demands has resulted in mining concessions covering 160 million ha, some 21%, of the total area of the Amazon basin (*Bebbington et al., 2018*). If these concessions are mined, they will have a devastating impact on primate populations and biodiversity.

*Gemstones and pearls*
The total accumulated value of exports of gemstones and pearls by the top exporting primate-range nations in 2016 was US$45 bn (Fig. 11C). India, Botswana, Thailand, Angola, and Namibia were top exporters of gemstones (Fig. 11C). China, the US, Belgium, and the UAE were major importers (Fig. 11D). Mining for gemstones such as diamonds, emeralds, and sapphires is accompanied by forest degradation, influxes of people, frontier settlements, road building, bushmeat hunting, and the disturbance of protected areas similar to that caused by the mining of metals and minerals (*Duffy, 2007*). Illegal gem mining coupled with logging in national parks poses a threat to the Endangered fork-marked dwarf lemur (*Phaner pallescens*) in Madagascar, and to the needle-clawed galagos (*Euoticus elegantulus*) in Cameroon (*Nomuh, 2018*). Illicit sapphire mining by itinerant miners in Madagascar has had a negative impact on the survivorship of lemurs (e.g., the Endangered ring-tailed lemur, *Lemur catta*), even in protected areas (*Duffy, 2007*; *Gould & Sauther, 2016*). Many primate range countries are among the most important

world producers of diamonds (e.g., Guyana, Brazil, Sierra Leone, Botswana, Tanzania), emeralds (e.g., Brazil, Zambia, Pakistan), sapphires (e.g., Pakistan, Sri Lanka, Madagascar), rubies (e.g., Pakistan, Myanmar, Thailand), jade (e.g., Myanmar), and other colored gemstones (*Shortell & Irwin, 2017*). Only a small fraction of the cutting, polishing, and processing of gemstones are conducted in the countries where this resource is extracted (*Shortell & Irwin, 2017*). The world production and trading of gemstones is dominated by a small set of international corporations including Rockwell Diamonds, Gem Diamonds, Lucara, Rio Tinto, Petra Diamonds, and De Beers (https://www.petragems.com/education/top-ten-diamond-companies-in-the-world-/), and monies generated by the gem trade are rarely used to support economic development in primate-range countries (*Howard, 2016*).

### Top primate-range, commodity-exporter countries, and top importer countries of these commodities

Our examination of the 2016 trade flows of the eight major commodity groups indicates that 55 primate habitat countries were exporters, and 42 countries were importers (Figs. 10 and 11; Tables S3 and S4). Among the top 10 exporting countries, Brazil, India, Indonesia, Malaysia, and South Africa accounted for 50% of the accumulated value of commodity exports in 2016 (Fig. 12A; Table S3). Just 10 importer countries accounted for 95% of the imported value (US$455 bn) of commodities from those 55 primate-habitat countries (Fig. 12B; Table S4). Among the major importers in 2016, China and the US combined, accounted for 58% of the imported value of these commodities. However, China was by far the single major importing nation, purchasing over twice the amount of commodities purchased by the US (Fig. 12B). In this regard, along with the US, China's emergence as an economic superpower has resulted in a devastating environmental footprint, that is, driving deforestation and biodiversity loss in primate-habitat countries (*Ascensão et al., 2018*).

### Revenues generated by commodity-exporting, primate-range countries in 2016

In 2016, US$4.4 trillion of natural resource commodities were traded by the world's countries (resourcetrade.Earth, 2018) and 25% (US$ 1.1 trillion) were commodities traded by countries in primate-range regions. The Neotropics accounted for 41% of the total value of these exports, Southeast Asia for 27%, Africa for 21%, and South Asia for 11% (Fig. 13A). Trade data from resourcetrade.Earth indicate that these export revenues increased significantly between 2000 and 2016 (Fig. 13A). International trade is a critical source of revenue for primate-range countries and, assuming that an increase in the GDPPC has a positive effect on food security, political stability, education, and health, we expect that it would favor primate conservation. However, primate range countries with a GDPPC of <10,000 US$ in 2018 had a higher percent of IUCN threatened-primate species than countries with a GDPPC of >10,000 US$ (Fig. 13B; Table S5).

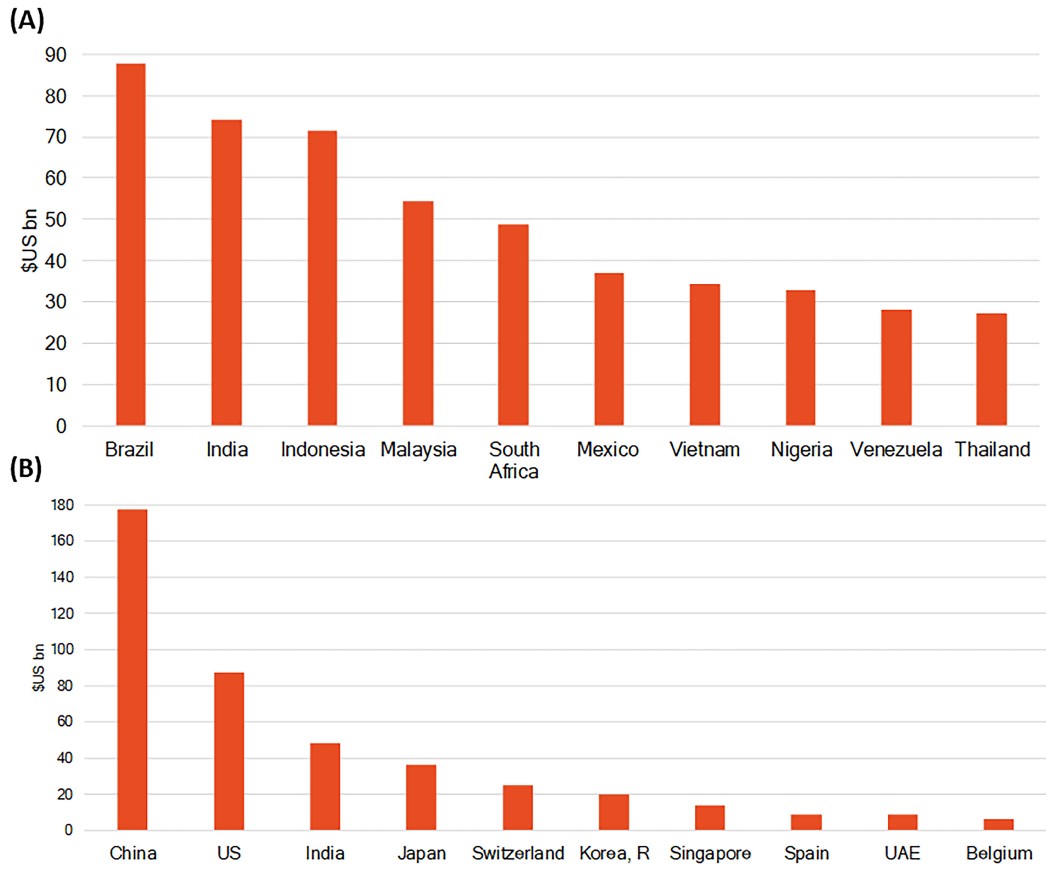

**Figure 12** (A) **Top 10 primate range exporter countries and (B) top 10 importer countries of forest-risk commodities in 2016.** These 10 exporting countries comprise all or part of the ranges of some 250 primate species (Table S5).

### Modeling the current and future (2050 and 2100) risk of primate species' extinction due to agriculture

As of 2016, the greatest number of primate species threatened with extinction (i.e., Vulnerable, Endangered, or Critically Endangered according to the IUCN Red List) are in Madagascar, Brazil, Indonesia, China, Vietnam, and Colombia (Table 2). In Madagascar and Southeast Asian countries such as Vietnam, Laos, and China, over 80% of the primates are threatened with extinction. Primate-habitat countries differ widely in the degree to which particular anthropogenic drivers have resulted in species decline (Table 2).

In Table S9, we aggregated 10 human land use patterns of the LUH2 dataset into three broad categories: forestry (defined as any human use of forested land), grazing (defined as pasture or rangeland for commodity production), and agriculture (defined as any crop production), for the top 15 primate-richest countries in the world. At the time of writing, forestry is the major threat to primate species in Peru, Cameroon, Indonesia, Laos, Malaysia, India, and Myanmar. In contrast, the conversion of forest into pasture for grazing, principally for cattle and pig production, is the major threat to primate species in Madagascar, Brazil, Colombia, and China. Among all land use patterns, agricultural activities are the greatest or second greatest threat to primate species in Indonesia, Brazil,

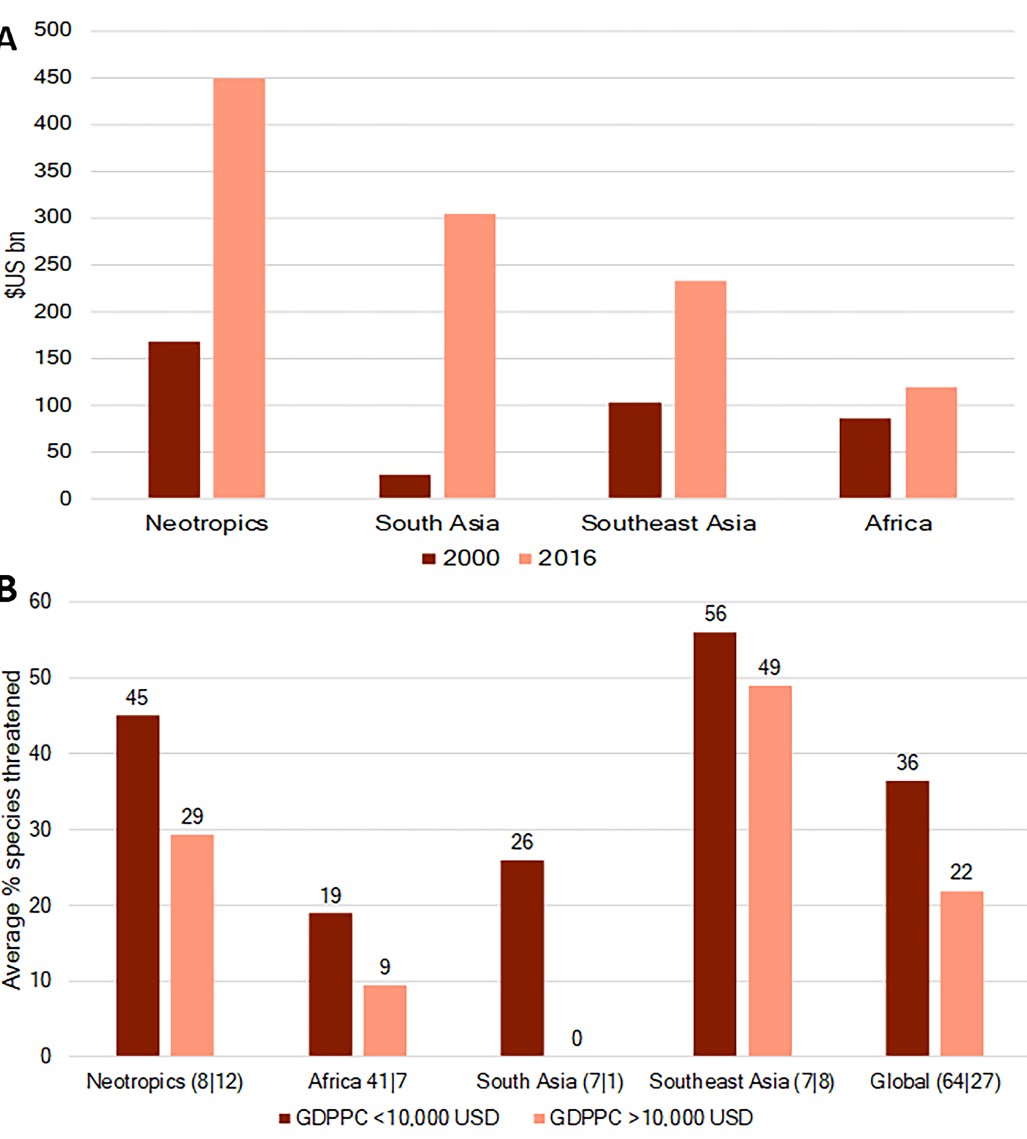

**Figure 13 (A) Estimated revenues generated by exports of natural-resource commodities by nations in primate-range regions in 2016.** (B) Average percent of primate species threatened and GDPPC US $10,000 for countries in primate range regions in 2016. The numbers in parenthesis next to the name of each region refer to the number of countries involved in each GDPPC class.

Vietnam, Laos, Malaysia, India, Nigeria, Myanmar, Thailand, Cameroon, and Cambodia (see Table S9 for all results per land use type).

The business-as-usual future scenario RCP 4.5 SSP-2 (*Fricko et al., 2017*) assumes continued economic development along historical patterns such that meat and food consumption converge slowly toward higher levels, trade is largely regionalized, and crop yields in low-income regions catch up with high-income nations. In the model, land use changes, however, are incompletely regulated, with continued deforestation (although at a declining rate) between 2016 and 2050. The model predicts that by the year 2050, each primate-habitat country is expected to see an increase in its number of

**Table 2 Top 15 countries with the greatest number of primate species threatened with extinction (2016).**

| Country | #Species | #Threatened | %Threatened | %Threatened due to forestry | %Threatened due to grazing | %Threatened due to agriculture |
|---|---|---|---|---|---|---|
| Brazil | 116 | 39 | 34 | 15 | 46 | 27 |
| Madagascar | 103 | 90 | 87 | 13 | 76 | 10 |
| Indonesia | 56 | 38 | 68 | 43 | 11 | 43 |
| Peru | 44 | 11 | 25 | 21 | 38 | 20 |
| Colombia | 37 | 18 | 49 | 21 | 64 | 11 |
| Cameroon | 32 | 9 | 28 | 38 | 18 | 37 |
| Nigeria | 26 | 11 | 42 | 27 | 31 | 35 |
| China | 25 | 20 | 80 | 30 | 46 | 18 |
| Vietnam | 22 | 19 | 86 | 37 | 6 | 52 |
| India | 22 | 12 | 55 | 44 | 5 | 41 |
| Malaysia | 20 | 14 | 70 | 52 | 4 | 40 |
| Laos | 18 | 15 | 83 | 51 | 7 | 37 |
| Thailand | 18 | 11 | 61 | 31 | 5 | 58 |
| Myanmar | 17 | 11 | 65 | 59 | 3 | 34 |
| Cambodia | 10 | 9 | 90 | 30 | 9 | 56 |

Notes:
The total number of primate species threatened with extinction (#threatened) are allocated to the negative effects of conversion of forested land for purposes of forestry, grazing, and agriculture (see *Chaudhary & Mooers, 2018* for additional details).
The number of primate species per country and IUCN status was taken from *Estrada et al. (2017)*, except for those of China which was taken from *Li et al. (2018)*.

threatened species (Table S8). This includes 13 newly threatened species in Madagascar (100% of species threatened) and 12 newly threatened primate species in Brazil (44% of species threatened) (Fig. 14). Land use changes by the end of the century are expected to result in six of the 15 primate-richest countries having 100% of their primate species threatened with extinction by the year 2100, and three additional countries having over 80% of their primate species threatened with extinction (Fig. 14; Table S8). On a more positive note, we found that the most ecofriendly climate change mitigation scenario, coupled with a sustainable socioeconomic trajectory RCP2.6 SSP-1 (*Van Vuuren et al., 2017*) which is defined as the world shifting toward a sustainable path characterized by healthy diets, low waste, reduced meat consumption, increasing crop yields, and reduced tropical deforestation, is expected to limit land use changes such that over the next 30 years (e.g., by 2050), no additional primate taxa will become threatened with extinction (Table S8). The scenario will require significant changes in human behavior and land use practices.

## TRADE AND FOOD SECURITY

Trade is a critical factor in food security and provides an income for approximately 30% of the world's active workforce (*Clapp, 2015*). While some countries lack the natural capital to produce enough food because they are restricted by land, climate, soil, political instability, or technology, other countries produce more food than they require. Open trade policies allow the free flow of food from countries with excesses to countries with deficits and, therefore, are expected to enhance world food security (*Lamy, 2011, 2012; Organisation*

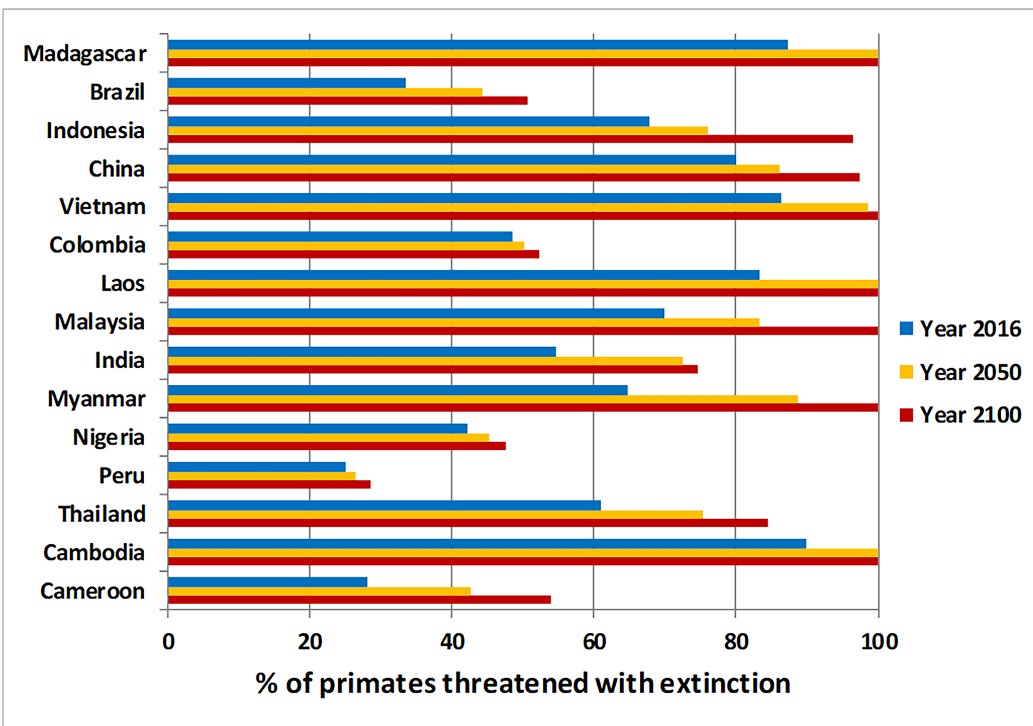

**Figure 14 Percent of primate species currently threatened with extinction (2016) and the percent projected to be threatened with extinction due to commodity driven land use changes by 2050 and 2100 under a business-as-usual scenario (RCP 4.5 SSP-2).** Data are presented only for the top 15 primate richest countries in the world. See Table S8 for the projected fraction of primate species threatened with extinction in 61 primate habitat country's under all six future scenarios and Table S9 for the contribution of each of the 10 human land use types to the total number of species threatened with extinction.

for Economic Co-operation and Development (OECD), 2013; World Bank, 2012). The Economist Intelligence Unit (2018) FSI ranges from 0 (lowest food security) to 100 (highest food security) (Text S1). In 2018, the mean value of the FSI, available for 43 of the 55 primate-range country commodity exporters was 49.0 (range 27.0 (Madagascar) to 69.2 (Argentina)) (Fig. 13A; Table S6). For the top 10 exporters, the mean FSI was 57.0 (range 38.0 (Nigeria) to 68.0 (Brazil)). In contrast, the mean value of the FSI for the top 10 commodity importer countries was 76.0 (range 50 (India) to 86 (Singapore)) (Fig. 15A; Table S7). Clearly, primate-range countries that are exporters of commodities lag behind importer nations in food security. These major importing nations tend to capture large volumes of export commodities from primate-range countries with limited positive effects for the exporting countries. This difference is underlined by the disparity in the 2017 GDPPC between these two groups of countries (Fig. 15B).

It is expected that trade helps food security by providing a safety net against oscillations in domestic food supply and by stabilizing prices (World Bank, 2015). Trade dependence, however, also means that countries are vulnerable to shifts in the trade policies of both food-importing and food-exporting nations, including tariffs and trade bans (Bloomberg, 2018; Cottrell et al., 2019). Importantly, the interconnectedness of global trade and

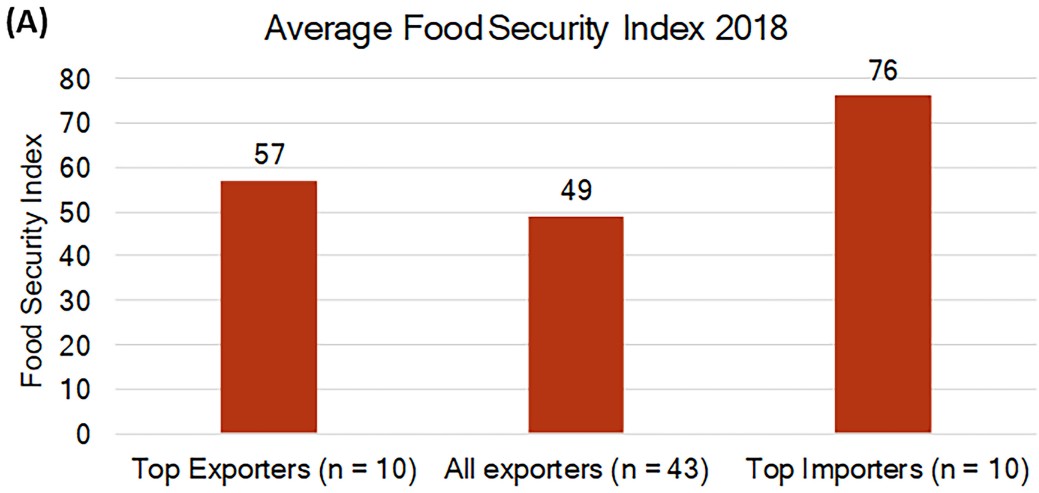

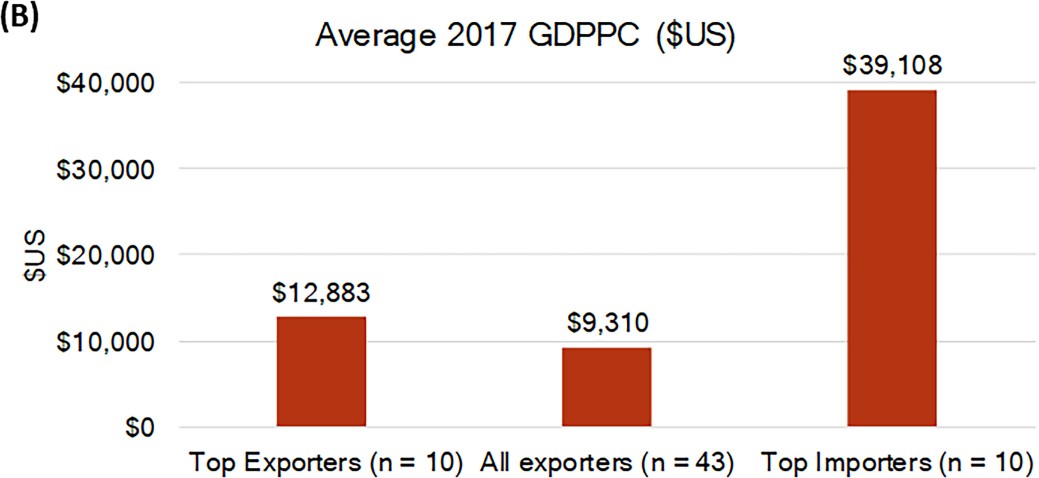

**Figure 15 (A) Food security index and (B) GDPPC in exporting and importing countries involved in forest-risk commodities trade.** We compare the average value of the food security index (FSI: 0 lowest to 100 highest) between top commodity exporting primate-range countries and top commodity importing countries in 2016. We also compare the average GDPPC for 2016 for the top primate-range commodity exporting countries and the top importer countries in 2016.

supply chains creates a risk that the storage and transport infrastructure that governs international trade might collapse (*Rudolff, 2015*)—such as when political and physical bottlenecks restrain the flow of food, water, and energy (*Bloomberg, 2018*). Political conflict is one of the main drivers of food insecurity, disrupting not only food production and distribution but also transport (*World Bank, 2015*). Shifts in market prices also have important consequences for consumers. For example, as China pays more for soybeans, whether in response to tariffs on US imports or to broaden its import profile, other consumers will pay more as well (*Bonato, 2017*). For low-income households around the globe that already spend most of their income on food, abrupt cost increases produced by higher import duties, or by unexpected shifts in trade flows, can have a significant negative impact on health, food security, the spread of disease, and political stability (*Rudolff, 2015*).

## CORRUPTION AND ILLEGAL FOREST CLEARING FOR COMMERCIAL AGRICULTURE

Factors that negatively impact human communities, also negatively impact policies and practices of environmental sustainability, and often result in habitat degradation and loss of biodiversity. One such factor is corruption. Information from the 2018 corruption perceptions index (CPI) of Transparency International (https://www.transparency.org) in which 0 = most corrupt and 100 = least corrupt, showed that commodity exporters in primate-range countries were, on average, more corrupt (average CPI of 33.1; range:18.0–68.0; Table S6) than importer countries (average CPI of 52.8; range 20.0–85.0; Table S7). Income inequality is often directly associated with corruption. For example, between 2001 and 2015 the richest 10% of Brazilians accounted for 61% of total economic growth (Oxfam, 2019a). Similarly, wealth inequality in Indonesia is so extreme that the four richest men in the country have more wealth than the combined total of the poorest 100 million people (Oxfam, 2019b). Both Indonesia and Brazil are ranked as the first and third primate-richest countries in the world, and increased corruption and income inequality is projected to result in high levels of environmental degradation, deforestation, and increased rates of primate extinctions (Estrada et al., 2018).

A recent report by Forest Trends indicates that since the start of the 21st century, illegal forest clearing for commercial agriculture and related exports has continued at an alarming rate in tropical regions (https://www.forest-trends.org/). This has resulted in high conservation costs. It is projected that unless governments ensure that forested land converted for production is acquired legally and sustainably, deforestation will continue to increase in regions where little commercial agriculture had previously existed, such as the Congo Basin (Lawsom et al., 2014). Governments that curb political freedom, free speech, a free press, and citizen rights and choices inevitably also have free license to engage in large-scale environmental destruction. The most recent (2019) evaluation by Freedom House, an independent watchdog group, shows that in many countries in Africa, South Asia, Southeast Asia, and the Neotropics, individual freedoms have declined precipitously in the last 13 years (Freedom House, 2019), and this is likely to result in a further decline in laws governing environmental and primate protection (Fig. 16).

## CONCLUSION AND CLOSING COMMENTS

The production of commodities such as soybeans, palm oil, natural rubber, and beef has direct and indirect impacts on biodiversity and primate population persistence because plantations are primarily large-scale, commercial monocultures that require complete clearing of natural vegetation. In addition, the use of pesticides and herbicides serve to eliminate remaining native biodiversity, and diminish the likelihood of habitat restoration (Fearnside, 2001). The extraction of hardwoods, minerals, land-based fossil fuels, and gemstones in undisturbed areas causes extensive forest degradation and forest loss. It also results in the construction of transportation networks and other infrastructure projects that open large and once remote forests to population migration, illegal logging, bushmeat hunting, and the illegal pet trade, critically impacting primate and other animal

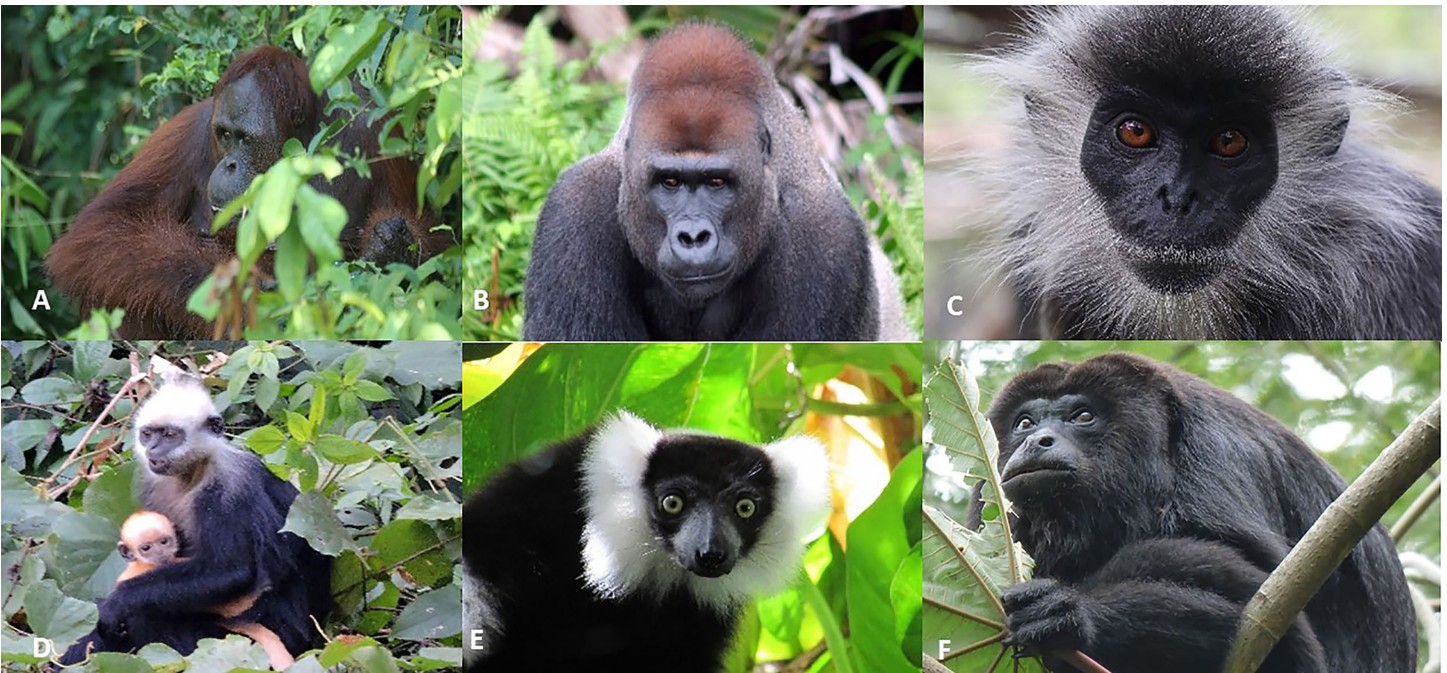

**Figure 16 Photos of selected primate species impacted by forest loss and degradation resulting from production of forest-risk commodities.** Also shown is their IUCN conservation status. Photo credits include the following: (A) Bornean orangutan (*Pongo pygameus*), Borneo. Status: CR (photo credit: R. Butler), (B) Western gorilla (*Gorilla gorilla*), Gabon. Status: CR (photo credit: R. Butler), (C) Indochinese Lutung (*Trachypithecus germaini*), Cambodia. Status: EN (photo credit: R. Butler), (D) white-headed langur (*Trachypithecus poliocephalus*, China. Status: CR (photo credit P. A. Garber), (E) Black-and-white ruffed lemur (*Varecia variegata*), Madagascar. Status: CR (photo credit: S. Johnson), (F) male black howler monkey (*Alouatta pigra*), Mexico. Status: EN (photo credit: S. Van Belle). EN, endangered; CR, critically endangered.

populations (*Alamgir et al., 2019*; *Latrubesse et al., 2017*; *Plumptre et al., 2015*; *Spira et al., 2017*; *Timpe & Kaplan, 2017*; *Winemiller et al., 2016*). Unfortunately, a projected massive expansion of transportation networks has been proposed as part of China's Belt and Roads Initiative. This will involve some 60 counties in Africa and Asia through a large-scale expansion of roads and rails in order to create a set of economic corridors designed to expand future global trade. This will result in permanent environmental damage, impacting primates and biodiversity by increasing agricultural production, the extraction of raw materials, as well as the construction of dams and other large infrastructure projects, causing habitat reduction and fragmentation, increasing human-primate conflict (*Ascensão et al., 2018*; *Liu et al., 2019*; *MERICS, 2019*) and contributing to climate change (*Barrett et al., 2013*).

Furthermore, global dietary changes toward greater meat consumption and greater dependence on vegetable oils as a result of improved living standards will encourage many primate-range countries to convert additional forested land into monocultures to meet national and global market demands (*Gouel & Houssein, 2017*; *Kastner et al., 2012*; *Tilman & Clark, 2014*). In order to achieve the goals of primate habitat conservation, it is imperative to decrease the world's demand for agricultural products (e.g., oil seeds, natural rubber, sugar cane) and the consumption of meat and dairy products (*Machovina,*

*Feeley & Ripple, 2015*; *Ranganathan et al., 2016*; *Stehfest et al., 2009*; *Willett et al., 2019*).
It is estimated that the world will need to convert an additional seven million ha of forested
land in the tropics to oil palm in the next few years to meet projected demand
increases (*Lawsom et al., 2014*). Per capita meat-consumption is growing globally and it is
anticipated to increase in the approaching decades with an expanding human population,
rising incomes in most regions of the world (*Henchion et al., 2014*), and the rapid
emergence of a global middle class (1.8–4.9 billion by 2030; *Kharas, 2017*). Trade evidence
suggests that land conversion for the production of forest-risk commodities has
expanded rapidly in primate-range regions, placing primate populations at greater risk
(Fig. 6; *Chaudhary & Brooks, 2017*). For example, a study that modeled agricultural
expansion and primate distributions for the end of the 21st century predicted that 68% of
the area currently occupied by primates will be turned over to agricultural use, affecting
75% of primate species worldwide (*Estrada et al., 2017*). A country level analysis under a
"worst-case" scenario (e.g., expected increase in conversion of forested land to agricultural
production) indicated that by the year 2100, the spatial distribution of primates in
Brazil will be reduced by 78%, 72% in Indonesia, 62% in Madagascar, and 32% in the DRC
(*Estrada et al., 2018*). Combined, these four countries account for approximately 64% of
all primate species. In a second study using the same modeling approach, the most
"optimistic" scenario (e.g., strong and effective legislation to protect primates and their
habitats) predicted a 51% reduction in the geographical range of primates in China by
the year 2100 (China has 25 primate species, nine of which are endemic) and an 87%
reduction under a "worse case" scenario (*Li et al., 2018*). Such extensive land conversion
will not only increase the susceptibility of primate species at risk of extinction, but it
will exacerbate the effects of climate change and cause severe disruption to local human
communities through extreme weather events such as increasingly severe storms, floods,
and droughts, leading to the high probability of a refugee crises (*Estrada et al., 2017*,
*2018*; *Henders et al., 2018*).

Unless a way is found to promote environmental protection by "greening" trade
(*Neumayer, 2001*; *Henders et al., 2018*), primate habitat loss and population decline will
continue unabated. Since the export of food and nonfood commodities may threaten
local food security, human safety, and political stability (*Food and Agriculture
Organization of the United Nations (FAO), 2019*), a balance needs to be achieved that
promotes a reduction in global market demands and takes into consideration the needs of
primate-range countries to develop their internal economies, to ensure food security for
their expanding human populations, and to protect their biodiversity (*Estrada et al., 2017*).
It has been suggested that corporations marketing products that contain tropical forest-
risk commodities should add environmental costs to products so that there is a continuous
renewal of resources dedicated to improving conservation and restoring natural habitats
(*Butler & Laurance, 2008*). From our analysis of forest-risk commodity trade data it is
notable that while 55 primate-range countries were identified as exporters of these
commodities in 2016, 95% of the exports were purchased by only 10 importing nations,
suggesting an unsustainable pattern of over-consumption, particularly by China and the
US. Action taken by importing countries, such as environmentally-friendly policies—for

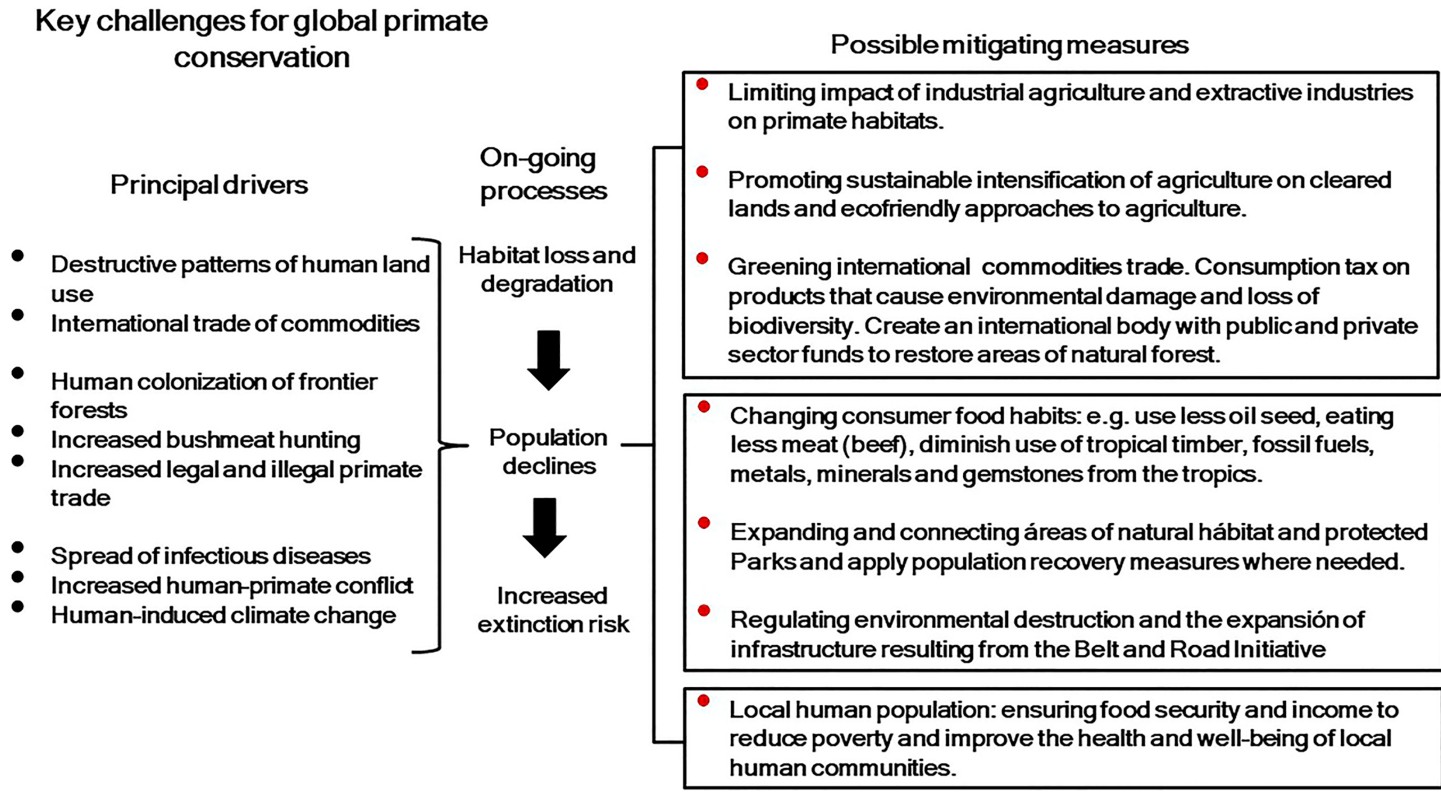

Key challenges for global primate conservation

Principal drivers

On-going processes

Possible mitigating measures

- Destructive patterns of human land use
- International trade of commodities
- Human colonization of frontier forests
- Increased bushmeat hunting
- Increased legal and illegal primate trade
- Spread of infectious diseases
- Increased human-primate conflict
- Human-induced climate change

Habitat loss and degradation

↓

Population declines

↓

Increased extinction risk

- Limiting impact of industrial agriculture and extractive industries on primate habitats.
- Promoting sustainable intensification of agriculture on cleared lands and ecofriendly approaches to agriculture.
- Greening international commodities trade. Consumption tax on products that cause environmental damage and loss of biodiversity. Create an international body with public and private sector funds to restore areas of natural forest.

- Changing consumer food habits: e.g. use less oil seed, eating less meat (beef), diminish use of tropical timber, fossil fuels, metals, minerals and gemstones from the tropics.
- Expanding and connecting áreas of natural hábitat and protected Parks and apply population recovery measures where needed.
- Regulating environmental destruction and the expansión of infrastructure resulting from the Belt and Road Initiative

- Local human population: ensuring food security and income to reduce poverty and improve the health and well-being of local human communities.

**Figure 17** Diagram summarizing key aspects of international commodities trade and primate conservation.

example, banning illegal timber purchases by the EU (*European Union, 2010*), the EU resolution on palm oil production and deforestation (*European Parliament, 2017*), and the Amsterdam Declaration to remove deforestation from food and nonfood commodity chains (*Amsterdam Declaration, 2015*)—embody important and constructive "green" changes that need to be adopted by the US, China, and other major importers of forest-risk commodity products from primate-range regions (Fig. 17).

Because demand for products by more affluent and more developed nations can lead to detrimental environmental pressures in primate-rich and resource-rich regions, countries that are major importers of agricultural and natural resource products from primate-range nations must become active sponsors of, and contributors to, habitat conservation efforts in exporting countries. Sponsoring an increase in land devoted to protected areas, which can improve human well-being for people living near them (*Naidoo et al., 2019*), contributing to improved conservation management, promoting community-managed forests, and strict adherence to laws and policies that integrate forest protection and commodity production, are all essential actions that can support biodiversity and the people who live in primate-habitat countries (*Porter-Bolland et al., 2012*; *Sharif & Saha, 2017*) (Fig. 13). Importing countries can actively support the regeneration of agricultural land and pasture to secondary forests as an important conservation measure (*Chaudhary & Brooks, 2017*). In short, a stronger worldwide effort at

regulating the negative impact of unsustainable commodity trade in primate-range regions is critically needed (*Henders et al., 2018*). Because the world's agricultural production and supply chains are controlled by a relatively small number of international corporations (e.g., Cargill, Monsanto, BASF, Dow, Syngenta, and DuPont) that are chartered in consumer nations that produce food in the global south and then exported to other countries for processing and consumption (*EcoNexus, 2013*), ethical responsibility needs to be borne by these corporations. The rise of industrial agriculture (including beef production) has been closely linked to the global seed and agrochemical industries, and this has led to a multitude of environmental problems including the degradation and destruction of ecosystems, a major decline in insect biodiversity, the expanded use of polluting and toxic agrochemicals, and the loss of agricultural biodiversity that threatens sustainable agriculture (*Clapp, 2017*, *2019*; *Sánchez-Bayo & Wyckhuys, 2019*). A similar situation has accrued over decades with the extraction and trade of fossil fuels, where a few corporations including ExxonMobil, Chevron, Shell, BP, ConocoPhillips, Peabody Energy, Consol Energy, and Arch Coal have controlled oil and gas production and distribution worldwide, and disproportionately contributed to the devastating effects of climate change (*Carbon Tracker, 2018*). The mining of metals and minerals also is controlled by a few companies among which the most dominant are Fresnillo, AngloAmerican, Newmont Mining, Barrick Gold, Coal India, and China Shenhua Energy (https://www.miningglobal.com/top10/top-10-mining-companies-world).

By 2050, the global population is projected to increase from 7.5 to 9.3 billion and estimates are that food production will need to increase from the current 8.4 billion tonnes to almost 13.5 billion tonnes a year (*Food and Agriculture Organization of the United Nations (FAO), 2014*). Much of this growth in the human population will take place in primate-range countries, where populations are projected to increasing from 5.1 to 7.3 billion (Fig. S7A in *Estrada et al., 2017*). A major sustainability challenge is to meet global nutritional needs through a healthier diet, and to provide food security for the world's human population while reducing the environmental impacts of the burning of fossil fuels, deforestation, desertification, climate change, and water and air pollution (*Davila & Dyball, 2018*; *Willett et al., 2019*). Achieving that level of production from a depleted natural resource base will require an expanded and accelerated effort to transition to sustainable agriculture to ensure world food security, to provide economic opportunities, and to protect the ecosystem services on which agriculture depends (*Food and Agriculture Organization of the United Nations (FAO), 2014*, *2017*, *2018a*, *2019*; *Willett et al., 2019*).

Although the goals of commodity-driven exporting countries may be to improve their economies and to meet the nutritional and socioeconomic needs of their populations, despite considerable year-to-year increases in revenue derived from agricultural and nonagricultural exports in primate-range regions, millions of their citizens remain poor, undernourished, and undereducated, and lack access to quality healthcare (*World Bank, 2018*; *Willett et al., 2019*). Given the rapid pace and large scale at which forests have been cleared in most primate-range countries, promoting "sustainable intensification" of agriculture on already cleared land would increase production and may help to forestall further land conversion (*Carlson et al., 2018*). In addition, connecting farmers who are

small landholders with international commercial agricultural entities, may also bring direct economic benefits to the rural poor, as long as their land is protected from debt or confiscation (*Goldsmith & Cohn, 2017*). However, intensification of agriculture that increases yields will not reduce global hunger as long as a small number of consumer nations distantly located from production areas continue to over-consume and waste food and other commodities. There needs to be a sustained global effort to increase food security in areas of the world where the hungry live, using eco-effective methods that encourage the sustainable productivity of multiple ecosystem services, reduce greenhouse gas emissions, and protect natural biodiversity (*Chaudhary & Mooers, 2018*; *Keating et al., 2010*; *Smith et al., 2019*). There is compelling evidence that greater powers of decision making by eco-friendly agricultural practices owned by small landholders who are sensitive to local markets and conditions, rather than profit maximizing large-scale industrial agribusiness, is key to food security in developing nations (*Runting et al., 2015*; *Tscharntke et al., 2012*). In reality, global commodity resource extraction is predicted to surge from 85 bn tons today to 186 bn by 2050 (https://resourcetrade.earth/) and this also will importantly contribute to critical losses of biodiversity, including primates, to the loss of ecosystem services and to huge volumes of GHGs, resulting from the destruction of valuable carbon sinks and from energy-intensive processing (*Chaudhary & Mooers, 2018*; *Willett et al., 2019*).

There is no doubt that additional research is required to examine the role of local and global market demands on primate conservation, including studies to evaluate the extent to which the reduction of land for purposes of agricultural conversion and nonfood resource extraction benefit local human and nonhuman primate communities. Given the crises that we face, applying economic tools to consumer nations, such as taxes on consumption and on agricultural resources, accompanied by investment in sustainable agri-environmental production and sustainable nonfood resource extraction, are viable alternatives to mitigate the negative impacts of global market demands and of the international commodity trade in primate habitats (*Kok et al., 2018*; *Larrosa, Carrasco & Milner-Gulland, 2016*). Our assessment of the international commodities trade and over-consumption by a small number of consumer nations suggests that conservation success requires a set of internationally agreed upon sustainable approaches to land productivity and to natural resource extraction to ensure food security, alleviate poverty, and mitigate forest loss and degradation (*Food and Agriculture Organization of the United Nations (FAO), 2019*). It also underlines the need for a stronger global resolve to regulate the negative impact of unsustainable commodities trade on primate habitats and biodiversity. Such global resolution needs to be matched by a reduction of the world's per capita demand for forest-risk food and nonfood commodities from primate-range regions. Primates and their habitats are a vital component of the world's natural heritage and culture. As our closest living relatives, nonhuman primates deserve our full attention, concern, and support for their conservation and survivorship.

## ACKNOWLEDGEMENTS

We thank Anthony B. Rylands for constructive comments on an early draft of the manuscript. Paul A. Garber is forever grateful to Jennifer A. Garber, Sara A. Garber, and

Chrissie McKenney for inspiring him to redirect his efforts to protecting the world's threatened primate populations. Alejandro Estrada is thankful to Erika and Alex for always supporting his interests in primate research and conservation. We are grateful to Dr. Sarah Bologna for her very useful edits on a final version of this article.

### Funding
Abhishek Chaudhary was funded by the Initiation Grant of Indian Institute of Technology (IIT) Kanpur, India (project number 2018386). The funders had no role in study design, data collection and analysis, decision to publish, or preparation of the manuscript.

### Grant Disclosure
The following grant information was disclosed by the authors:
Initiation Grant of Indian Institute of Technology (IIT) Kanpur, India: 2018386.

### Competing Interests
The authors declare that they have no competing interests.

### Author Contributions
- Alejandro Estrada conceived and designed the experiments, performed the experiments, analyzed the data, contributed reagents/materials/analysis tools, prepared figures and/or tables, authored or reviewed drafts of the paper, approved the final draft.
- Paul A. Garber conceived and designed the experiments, performed the experiments, analyzed the data, contributed reagents/materials/analysis tools, prepared figures and/or tables, authored or reviewed drafts of the paper, approved the final draft.
- Abhishek Chaudhary conceived and designed the experiments, performed the experiments, analyzed the data, contributed reagents/materials/analysis tools, prepared figures and/or tables, authored or reviewed drafts of the paper, approved the final draft.

### Data Availability
All data reported are presented in the Tables, Figures, and Supplemental Material. Data used in this article was gathered from public domain sources indicated in the section Survey Methodology and in other parts of the text.

### Supplemental Information
Supplemental information for this article can be found online at http://dx.doi.org/10.7717/peerj.7068#supplemental-information.

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
