# Peer review of "Expanding global commodities trade and consumption place the world's primates at risk of extinction"

_PeerJ, doi:10.7717/peerj.7068_

## Round 0.1 · original submission · Major Revisions

I received comments from reviewers. All of them suggested that your paper is an important contribution to the knowledge on the primate conservation of and fits well with PeerJ's goals. Nevertheless, they have also pointed out several improvements that need to be done before your paper could be accepted for publication. Hence, please carefully consider ALL points mentioned in the reviewers' comments, explain every change made, and provide proper rebuttals for any remarks not addressed. I look forward to receiving an updated version of your manuscript.

·

Basic reporting

The manuscript of Estrada et al. presents an assessment of the threat to primates as a result of the advance of international trade in commodities in 75 countries that hold the highest diversity for the group. Agricultural commodities (soy, rubber, livestock and vegetable oil) and non-agricultural commodities (oil and ore) were selected, with assessments being made for countries in the Neotropical region, Africa and Madagascar, South and Southeast Asia. The authors compiled data on the loss of forest cover between 2001 and 2015, data on international trade in commodities for the years 2000 and 2016, data on the Gross Domestic Product for the countries involved. Primates, used here as a response group, were evaluated by the number of currently threatened species and habitat loss scenarios associated with six different RCPs (Representative Concentration Pathway) for the years 2050 and 2100. Based on an equation of species-area relationship, the authors estimated the number of species that will be extinct in the future for each of the RCPs.

Experimental design

In general, the manuscript is very interesting, well written and with many relevant data on the dynamics of the world trade of commodities. My main point concerning the study was the weak association between commodity production and threats to primates, as suggested by the title of the manuscript. It is quite intuitive to assume that species of primates present in commodity-producing countries, whose activities demand more and more areas, would be increasingly threatened. But the authors failed to show this relationship. All the dynamics of the trade and expansion of commodities were well demonstrated, but it was not demonstrated if the conservation status of the primates was changed during the considered period. This relationship could indicate whether the expansion of commodity production would actually pose a more significant threat to primates. The ideal would be to demonstrate the variation of the conservation status of the species within the analyzed period for the change of the commodities. Perhaps the calculation of the Red List Index variation (Bubb et al. 2009) is a good way to indicate if the conservation status of species has changed along with the loss of forest cover and commodity expansion. Thus, a simple graph showing in the Y-axis the variation of the RLI between 2000 and 2016 and in the X-axis the variation of the commodity production for each primate-range country would give an idea of this relation.

The main point raised by the authors is that international trade in commodities generates large environmental impacts in exporting countries. The use of the primate group as an example is only circumstantial since virtually any taxonomic group could be used in this context. So my recommendation would be to change the order of subjects in the introduction so that the text begins by talking about commodities, the characteristics of primate-range countries and then about the expected impacts on primates.

Another critical issue is that some regions of Brazil, such as the Atlantic Forest, are not producing commodities for export, but they have many species of primates, including endangered species. In the opposite direction, there is the Cerrado, which has low primate diversity but is the main export region of agricultural commodities in the country. Argentina would be the equivalent of the Cerrado, that is, it has a low diversity of primates (only 6 species) and high production of commodities. Thus subregional situations may mask the results, and this should be better discussed in the text.
Analysis of future scenarios for primates is unclear or not adequately presented. The authors used land use projections in different climatic scenarios to indicate the area available for primate occurrence. The result of the analysis suggests that there may be a significant increase in the number of endangered species, but a species-area relationship would indicate the number of extinct species, i.e., the difference between current and future species richness. Nevertheless, the authors indicated in Table S8 that 100% of all primates would be threatened with extinction in most of the scenarios. Does it mean that the species-area equation did not indicate any extinction? In my opinion, the table should show the number of species that might be extinct in the future due to land-change and climate change.
Based on the results, the authors suggest some measures that could be taken to mitigate the impacts. Some of the proposals (eg, "sustainable intensification," "reduce greenhouse gas emissions" or "protect natural biodiversity") are very generic and could be applied to any taxonomic group that is being impacted by the production of commodities. The most relevant would be to point out specific actions for species of primates that may disappear shortly.

Some specific remarks:

The introduction presented is quite similar to the introduction of Estrada et al. (2017), which, in my opinion, represent a self-plagiarism.

• “Nonhuman primates (primates from here on), our closest biological relatives, are an essential component of tropical biodiversity, contributing to forest regeneration and ecosystem health. Primates play important roles in the livelihoods, cultures, and religions of many societies and offer unique insights into human evolution, biology, behavior, and the threat of emerging diseases.” (lines 47-51)

• “Nonhuman primates (primates hereafter) are of central importance to tropical biodiversity and to many ecosystem functions, processes, and services. They are our closest living biological relatives, offering critical insights into human evolution, biology, and behavior and playing important roles in the livelihoods, cultures, and religions of many societies.” (Estrada et al. 2017).

The authors should re-write this part, and also avoiding the exceeding self-citation. I counted 23 times that Estrada et al. 2017 and Estrada et al. 2018 were cited in the text. Citations of papers from the second author (Chaudhary & Brooks, 2017 and Chaudhary & Mooers, 2018) sums 11 times. Altogether, the 34 citations of paper from the authors are 11 times more frequent than the citation of other papers.

Still talking about citations and references, there are several errors along the manuscript that should be corrected. Here is a list of the mistakes:

Line 54 . IUCN Red List would be a better source to be cited here, since the organization is the one responsible for the definition of threatened status of the species.
Line 92 – Check the citation format of Henders, Persson & Kastnter 2015. Should be the same of Henders et al. 2015, as in Line 72.
Line 140 – Missing reference: Clapp 2016 and FAO 2015.
Line 140 – Which FAO 2018 is this? 2018a or 2018b?
Line 277 – Please correct Chaudhary and Brookes, 2017 to Chaudhary and Brooks, 2017
Line 315 – Keep the consistency along the text: either cite 3 authors or use et al. (as in Henders et al. 2015 or Lanjouw et al. 2015)
Line 318 – Missing reference: IUCN 2018 (Is this Grasp & IUCN 2018?).
Line 441 – Lewis, Edwards & Galbraith 2015: keep the consistency along the text: either cite 3 authors or use et al. (as in Henders et al. 2015 or Lanjouw et al. 2015)
Line 535 – Change Nomuh Forbanka 2018 to Nomuh 2018.
Line 725 – Missing reference: FAO, IFAD & WFP 2015.
Line 751 – Is this Porter-Bolland et al. 2016 or 2012, as in the reference list?
Line 784 – Which FAO 2018 is this? 2018a or 2018b?
Line 791 – Missing reference: World Bank 2018.
Line 14 (Table 1) – Missing reference: Curtis et al. 2015.
Line 866 - Please check the reference Ahrends et al. 2015, which is not cited in the text.
Line 898 - Please check the reference Barrett et al. 2013, which is not cited on the text.
Line 938 – Put Clapp 2019 after Clapp 2017.
Line 948 - Please check the reference Cottrell et al. 2019, which is not cited in the text.
Line 968 - Please check the reference Diesmos et al. 2014, which is not cited in the text.
Line 1083 - Please check the reference Harfoot et al. 2017, which is not cited in the text.
Line 1108 – Put IUCN 2019 after IUCN 2015.
Line 1108 – Please check the reference IUCN 2019, which is not cited in the text. Is this IUCN Redlist 2019 cited in Figure 1?
Line 1126 - Please check the reference Kok et al. 2017, which is not cited in the text.
Line 1173 – Reference duplicated (the same as in line 1165)
Line 1178 - Please check the reference Liu et al. 2019, which is not cited in the text.
Line 1191 – Incomplete reference? It is apparently cited as Mazumder et al. 2014 in line 371.
Line 1203 - Please check the reference Meyfroidt et al. 2013, which is not cited in the text.
Line 1225 - Please check the reference Nantha & Tisdell 2009, which is not cited in the text.
Line 1228 - Please check the reference Nepstad et al 2006, which is not cited on the text.
Line 1264 - Please check the reference Ranganathan et al. 2016, which is not cited in the text.
Line 1280 - Please check the reference Seto et al. 2012, which is not cited in the text.
Line 1362 - Please check the reference Zalles et al. 2019, which is not cited in the text.

Table 2. Please choose a column to order the data. I suggest ordering descending by # of species.

Validity of the findings

Please see the previous item.

Additional comments

Please see the previous item.

·

Basic reporting

This is an excellent and much needed analysis of how commodities production, trade, and consumption patterns need to change in order to prevent the extinction of primates and their habitats. The analysis is rigorous and relevant and it is clear that the authors have produced a publication-ready manuscript and I thank the authors for working so hard on that. Overall, I think this is a critically important manuscript to be published at this time and I believe your journal should do whatever you can to promote the piece once it is published.

I have just a couple of comments for the authors to consider for this and future articles.

Experimental design

no comment

Validity of the findings

no comment

Additional comments

These are those two things to consider:

1) I know that we primatologists like to think in the four-five region system for grouping data. Lately, I have been wondering how useful this grouping process is for understanding patterns. I refer to the region designations: Neotropical, Africa, South Asia, and SE Asia. I know these terms come from tradition and the desire for standardization and gross categories. First, I don’t know that the categories really tell us anything. Second, I am learning from our students that the term New World and Neotropical (because it is the same thing) is offensive because it erases indigenous history in the Americas. Third, the comparisons are uneven. In one terms, we are using a forest/vegetation term and in the others, continent terms. If you are stuck on the four big categories, perhaps the Americas. I, however, believe the countries are much more important for the actual outcomes here.
2) Figure 17 is important. I wonder if there is a way to do this in a more meaningful way to address how individual consumption and behavior influences larger-scale systems. I believe the bottom boxes could be more creatively integrated to influence greater impact. Perhaps bring in the local, regional, global scale here.

Here are some minor things:

Line 78: what are some examples of the negative conservation consequences?
Line 87: perhaps define forest-risk commodities. Examples are provided but what places something in this category? Large scale monoculture?
Line 135: Was 2016 used as most recent example with relatively complete data? What were 2000 and 2016 chosen? I think I know why but it might good to be explicit.
Line 232: Change ‘amount’ to ‘degree’
Line 271: Add comma after SE Asia and change ‘he’ to ‘the’. This sentence is a little awkward with the ‘also…’
Line 316: Make soybeans(s) plural to be consistent with use in other places.
Line 750: Are primate-range and primate-habitat interchangeable? I noticed range used in other areas. I know that they are interchangeable but do you want to use them both?
Line 792: What is a native forest and why is it important?

---

## Round 0.2 · accepted · Accept

Both referees reviewed your updated version, and they are happy with the updated version of your paper. Congratulations! Please work with our production team so that your article can be published as soon as possible.

·

Basic reporting

After reading the new version of the manuscript, I am quite confident that the text is now in shape to be published. The authors had revised all critical points raised on the first revision. Therefore, I have no objection to its publication.

Experimental design

See above.

Validity of the findings

See above.

Additional comments

See above.